# Task-induced functional brain connectivity mediates the relationship between striatal D2/3 receptors and working memory

Matthew M Nour[1,2,3,4,5]*, Tarik Dahoun[2,3,6], Robert A McCutcheon[1,2], Rick A Adams[7,8], Matthew B Wall[9], Oliver D Howes[1,2,3]

[1]Institute of Psychiatry, Psychology and Neuroscience (IOPPN), King's College London, London, United Kingdom; [2]MRC London Institute of Medical Sciences (LMS), Hammersmith Hospital, London, United Kingdom; [3]Institute of Clinical Sciences, Imperial College London, London, United Kingdom; [4]Max Planck UCL Centre for Computational Psychiatry and Ageing Research, University College London, London, United Kingdom; [5]Wellcome Centre for Human Neuroimaging (WCHN), University College London, London, United Kingdom; [6]Department of Psychiatry, University of Oxford, Oxford, United Kingdom; [7]Institute of Cognitive Neuroscience (ICN), University College London, London, United Kingdom; [8]Division of Psychiatry, University College London, London, United Kingdom; [9]Imanova Centre for Imaging Sciences (Invicro Ltd), Hammersmith Hospital, London, United Kingdom

**Abstract** Working memory performance is thought to depend on both striatal dopamine 2/3 receptors (D2/3Rs) and task-induced functional organisation in key cortical brain networks. Here, we combine functional magnetic resonance imaging and D2/3R positron emission tomography in 51 healthy volunteers, to investigate the relationship between working memory performance, task-induced default mode network (DMN) functional connectivity changes, and striatal D2/3R availability. Increasing working memory load was associated with reduced DMN functional connectivity, which was itself associated with poorer task performance. Crucially, the magnitude of the DMN connectivity reduction correlated with striatal D2/3R availability, particularly in the caudate, and this relationship mediated the relationship between striatal D2/3R availability and task performance. These results inform our understanding of natural variation in working memory performance, and have implications for understanding age-related cognitive decline and cognitive impairments in neuropsychiatric disorders where dopamine signalling is altered.

DOI: https://doi.org/10.7554/eLife.45045.001

**\*For correspondence:**
matthew.nour.18@ucl.ac.uk

## Introduction

Working memory is the short-term maintenance, evaluation and manipulation of information 'online', necessary for higher cognitive processing (*D'Esposito and Postle, 2015*; *Millan et al., 2012*). Working memory impairment is an important component of age-related cognitive decline, (*Braskie et al., 2008*; *Dahlin et al., 2008*) and is reliably present in multiple neuropsychiatric disorders, where it predicts poor functional outcomes (*Millan et al., 2012*). The neurobiology underlying working memory variability in health and disease remains unknown, however converging evidence implicates the neuromodulator dopamine (*Cools and D'Esposito, 2011*). Moreover, age-related decline in striatal

dopamine signalling could contribute to the cognitive decline seen in normal ageing, (*Braskie et al., 2008*; *Dahlin et al., 2008*; *Bäckman et al., 2000*; *Berry et al., 2018a*; *Berry et al., 2016*; *Matuskey et al., 2016*) and altered striatal dopamine signalling in schizophrenia and Parkinson's Disease could contribute to cognitive impairments in these disorders (*Millan et al., 2012*; *Simpson et al., 2010*; *McCutcheon et al., 2018a*; *Meder et al., 2019*).

One influential theory postulates that striatal dopamine 2/3 receptors (D2/3R) gate the flow of new information into the cortex, setting the balance between working memory stabilisation and updating (*Dahlin et al., 2008*; *Cools and D'Esposito, 2011*; *Bäckman et al., 2011*; *Frank et al., 2001*; *Frank and O'Reilly, 2006*). Recent animal studies show that striatal D2Rs induce changes in cortical activity and task-related mesocortical synchrony. Specifically, postsynaptic striatal D2R over-expression in mice causes a deficit in inhibitory signalling in the prefrontal cortex (PFC), (*Kellendonk et al., 2006*; *Li et al., 2011*) reduced activity in ventral tegmental area (VTA) dopamine neurons, and reduced VTA-PFC synchrony during working memory (*Krabbe et al., 2015*; *Duvarci et al., 2018*). These changes are accompanied by impairments in cognitive tasks that recruit the prefrontal cortex, including those that involve working memory (*Kellendonk et al., 2006*; *Duvarci et al., 2018*; *Simpson and Kellendonk, 2017*). Human and non-human primate studies also implicate the (dorsal) caudate in working memory function (*Dahlin et al., 2008*; *Rieckmann et al., 2011*; *Levy et al., 1997*).

In humans, working memory-induced changes to whole-brain functional organisation can be studied using functional magnetic resonance imaging (fMRI), which allows measurement of the functional connectivity between cortical brain regions (*Shine and Poldrack, 2018*; *Shine et al., 2016*). Functional connectivity in this context refers to the pairwise correlation between the time course of neural activity in each region, and is a marker of shared information processing (*Rubinov and Sporns, 2010*). Cortical brain regions can be clustered into 'networks' based on their pattern of functional connectivity during rest (*Rubinov and Sporns, 2010*). Recent human fMRI studies indicate that working memory performance is supported by changes in the functional connectivity and organisation of cortical networks. Specifically, working memory performance induces a decrease in functional connectivity within the default mode network (DMN) (*Cole et al., 2014*; *Finc et al., 2017*; *Liang et al., 2016*). The DMN is the prototypical 'task negative network' that typically shows reductions in activity during working memory tasks, and includes ventromedial prefrontal cortex and posterior cingulate cortex (*Finc et al., 2017*). Moreover, task-induced connectivity changes within the DMN predict performance on working memory tasks such as the n-back (*Finc et al., 2017*). Working memory performance is also accompanied by changes in functional connectivity within 'task positive networks' (TPN) such as the dorsal attention network (DAN) and the frontoparietal network (FPN), which typically show increases in activity during working memory tasks (*Shine and Poldrack, 2018*).

Together, these findings suggest that striatal D2/3R levels, particularly within the caudate, might exert an influence on working memory performance by modulating task-induced functional connectivity within task-relevant cortical networks. However, this hypothesis has not been tested in humans. In this study we address this question directly in a sample of 51 healthy volunteer participants. We used fMRI to measure functional connectivity within the DMN and TPN during a letter n-back working memory task that required updating of working memory representations. In the same participants, we measured baseline striatal D2/3R availability using positron emission tomography (PET). Our primary hypothesis was that striatal D2/3R availability would directly correlate with the magnitude of task-induced connectivity changes within our studied cortical networks, and that this relationship would mediate the influence of striatal D2/3Rs on performance.

## Results

### Task design, behaviour and task-induced neural activation

All participants completed 18 task blocks of a letter n-back working memory task (six blocks each of 0-, 1- and 2-back conditions, in pseudo-randomised order) during fMRI scanning (*Figure 1a*). During the task, participants indicated whether the 'target' letter was present or absent on each trial, as quickly as possible. In the 0-back condition the target letter was specified at the start of the task block and remained unchanged throughout the block. In the 1-back and 2-back conditions the target

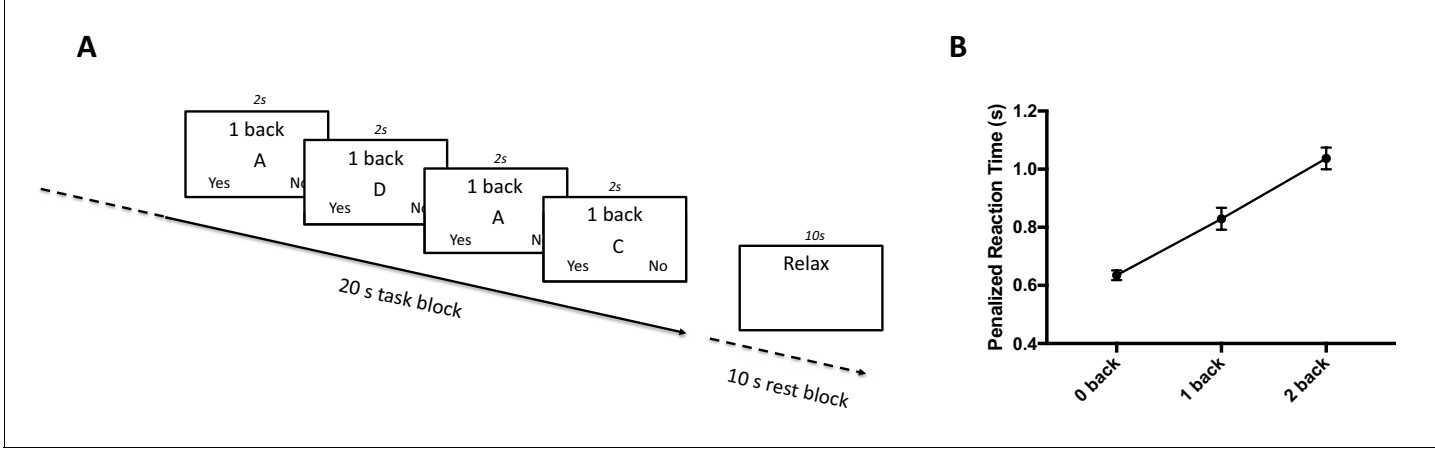

**Figure 1.** Task design and behavioural results. (**A**) Letter n-back task design. The task consisted of 18 task blocks (20s each, interspersed with 10s rest blocks) that were pseudo-randomised to 0-, 1-, or 2-back conditions (6 blocks of each condition per task session). Participants indicated, as quickly as possible, whether the target was present or absent at the appearance of each letter, using two buttons of an MR-compatible button box. Each letter appeared on the screen for 2 s before the next letter was shown. This figure shows the last 4 trials of a 1-back task block, followed by a rest block. (**B**) Task behaviour. Penalised reaction time ($pRT$, using penalization ratio of 2.5) increases (indicating poorer performance) with increasing working memory load. Plotted as mean ± S.E.M.

DOI: https://doi.org/10.7554/eLife.45045.002

letter changed on every trial (i.e. target letter for trial $t$ was the letter presented on trial $t$-1 for the 1-back condition, and trial $t$-2 for the 2-back condition).

We used a performance metric that is sensitive to processing speed and response omissions, as well as false positive responses. One such metric is the penalized reaction time ($pRT$, where reaction time set to maximum response time (2s) for incorrect or omitted responses) (**Finc et al., 2017**). However, using a fixed penalization of 2s fails to take into account individual- and condition-specific differences in mean reaction time. Consequently, we used a modified $pRT$ metric, in which the reaction time of an incorrect response was replaced by a $pRT$ that was a fixed ratio of the subject- and condition-specific $RT$. For clarity, we report behavioural results using the $pRT$ defined with a penalization ratio of 2.5 (hereafter, $pRT(2.5)$), however the statistical significance of all results is unchanged when using penalization ratios from 2.5 to 4 (hereafter $pRT(2.5-4)$).

We used this proportional $pRT$ measure to derive a single measure of working memory 'robustness' to increasing working memory load. This was defined as the negative regression coefficient relating $pRT$ to memory load ($-\Delta pRT$). A value of 0 indicates that performance is unaffected by the increase in task difficulty, and increasingly negative values indicate that performance degrades with increased cognitive demands.

Performance ($pRT(2.5)$) was negatively related to working memory load, indicating that our performance metric is sensitive to increasing cognitive demands (repeated measures ANOVA indicates a mean effect of cognitive load: $F_{2,100} = 68.7$, $P < 0.001$. Post-hoc paired t-tests indicate a significant difference between all pairwise comparison: t-tests: all $t_{50} > 5.3$ and $P < 0.001$) (**Figure 1b**).

As expected, (**Owen et al., 2005**) in a standard fMRI activation analysis there was activation in a widespread frontoparietal network that was parametrically related to working memory load, and deactivation in a network involving the ventromedial prefrontal cortex and posterior cingulate cortex (at $P < 0.05$, following whole-brain family wise error correction). Within the striatum, the activation cluster extended primarily into the dorsal striatum, in line with the postulated role of this region in updating working memory (**Dahlin et al., 2008**) (see **Figure 2a** and **Table 1** for activation/deactivation clusters). There was no significant linear or quadratic relationship between neural activation/deactivation and behavioural performance ($-\Delta pRT$ (2.5)) when this variable was used as a regressor in a second-level voxel-wise analysis.

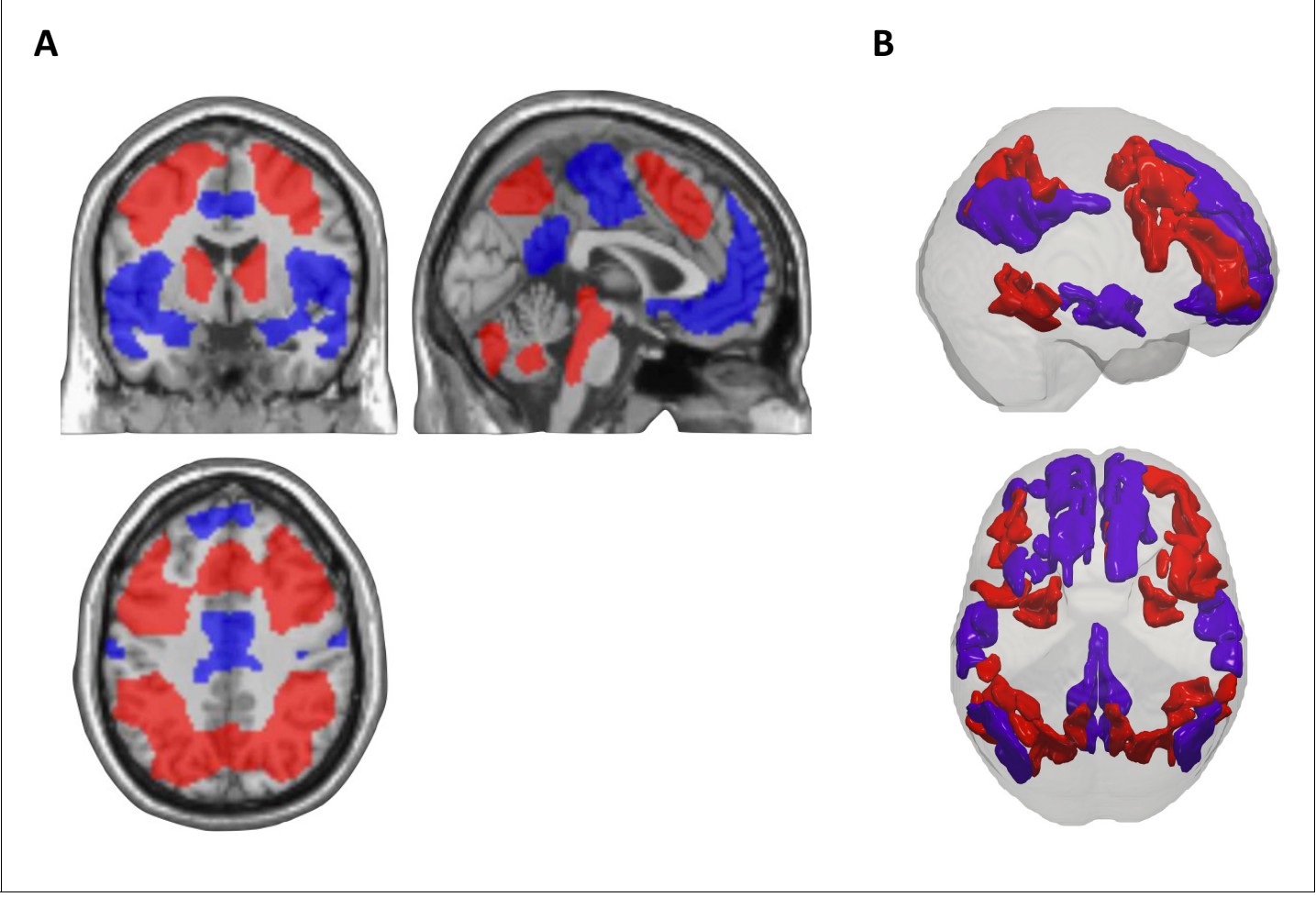

**Figure 2.** Task-related neural activation and task-related cortical networks. (**A**) Significant task-induced activations (red) and deactivations (blue) that correlate with increasing working memory load, showing increased activation within dorsolateral prefrontal cortex and parietal cortex, and decreased activation within ventromedial prefrontal cortex, posterior cingulate cortex and medial temporal lobe. Activation and deactivation maps thresholded at whole-brain cluster threshold (cluster-level family-wise error corrected P < 0.05), and displayed in coronal, sagittal and axial sections overlaid on a single-subject normalized T1 image in MNI space (see *Table 1* for statistical results). (**B**) Lateral (top) and top-down (bottom) rendering illustrating nodes of the empirical task-positive (TPN, in red) and default mode (DMN, in blue) networks identified by the community detection algorithm.
DOI: https://doi.org/10.7554/eLife.45045.004

## Task-induced functional connectivity change

To investigate task-induced functional connectivity changes within task-relevant cortical networks, we first used a data-driven community detection algorithm (*Blondel et al., 2008*; *Lancichinetti and Fortunato, 2012*) to partition the cortex into group-level task positive (TPN, 53 nodes) and default mode (DMN, 42 nodes) networks, using the fMRI signal time course during the baseline (0-back) task condition (*Figure 2b*). The 'empirical' TPN and DMN networks derived from this algorithm overlapped respectively with the spatial activation and deactivation maps from the standard fMRI activation analysis, serving to highlight the functional relevance of these networks. Specifically, the set of voxels defined by the DMN nodes overlapped primarily with the voxels of the cortical deactivation cluster map (Dice similarity coefficient (DSC) = 0.22), compared to the voxels comprising the cortical activation clusters (DSC = 0.08). Conversely, the set of voxels defined by the empirical TPN nodes overlapped with the cortical activation cluster map (DSC = 0.33), but not cortical deactivation cluster map (DSC <0.001). Moreover, the empirical DMN node assignments overlapped with the *a priori* network assignments from the Gordon cortical parcellation for the DMN (DSC = 0.92) but not with the *a priori* Gordon assignments for the task-positive networks ('DAN and FPN') (DSC = 0.08).

**Table 1.** Whole brain activation/deactivation results for parametric working memory load regressor.

| | Peak | | MNI coordinates (mm) | | |
|---|---|---|---|---|---|
| | P(FWE-corr) | T | X | Y | Z |
| **Activation clusters** | | | | | |
| Left middle/superior frontal gyrus | <0.001 | −30 | −30 | 4 | 56 |
| Right middle/superior frontal gyrus | <0.001 | 26 | 26 | 4 | 57 |
| Left middle frontal gyrus | <0.001 | −39 | −39 | 6 | 33 |
| Left precuneus/superior parietal lobule | <0.001 | -8 | -8 | −69 | 51 |
| Right precuneus/superior parietal lobule | <0.001 | 9 | 9 | −63 | 54 |
| Right superior parietal lobule | <0.001 | 33 | 33 | −48 | 44 |
| **Deactivation clusters** | | | | | |
| Left central operculum/posterior insula | <0.001 | 10.35 | −36 | −16 | 18 |
| Right central operculum/posterior insula | <0.001 | 10.24 | 40 | −16 | 22 |
| Left central operculum/anterior insula | <0.001 | 9.30 | −36 | 3 | 15 |
| Left posterior cingulate | <0.001 | 7.79 | -4 | −48 | 27 |
| Right posterior cingulate | 0.012 | 5.63 | 6 | −48 | 21 |
| Right posterior cingulate | 0.455 | 4.26 | 15 | −44 | 6 |

Activation/deactivation peaks present in the significant clusters at whole-brain threshold of P < 0.05 (family wise error (FWE) – corrected), using a cluster defining threshold P < 0.001 (uncorrected) for both contrasts. Anatomical labelling corresponds to the peak MNI co-ordinate. MNI = Montreal neurological institute.

DOI: https://doi.org/10.7554/eLife.45045.003

Conversely, the TPN nodes showed the opposite pattern, overlapping with the Gordon 'DAN and FPN' (DSC = 0.94) but not DMN (DSC = 0.04).

For each participant we defined the task-induced functional connectivity change within the DMN, TPN and the edges connecting the DMN and TPN (DMN-TPN) as the mean regression coefficient ($w_1$) of a linear model describing functional connectivity strength (Fisher z-transformed r-value) of each edge (dependent variable) as a function of cognitive load (independent variable). Task-induced functional connectivity significantly decreased with working memory load in the DMN ($w_1$ significantly less than zero: $t_{50}$ = -2.33, P = 0.02, one-sample t-test), and significantly increased with working memory load in the TPN ($w_1$ significantly more than zero: $t_{50}$ = 2.70, P = 0.01, one-sample t-test) (*Figure 3a and b*).

## Relationship between functional connectivity change and striatal D2/3Rs availability

[11C]-(+)-PHNO binds selectively to both D2 and D3 receptors, and the D3R fraction of this measure differs between the ventral and dorsal striatum. In the (dorsal) caudate, the D3R fraction of the [11C]-(+)-PHNO $BP_{ND}$ signal is negligible, while in the ventral striatum (including accumbens) the D3R fraction has been estimated at 20–25% (*Tziortzi et al., 2011*). In addition to molecular differences between striatal sub-regions, functional differences also exist. Specifically, working memory has been proposed to be supported by the (dorsal) caudate, with the accumbens being more relevant for limbic and reward processing (*Dahlin et al., 2008*; *Rieckmann et al., 2011*; *Levy et al., 1997*; *McCutcheon et al., 2019*; *Haber, 2003*). Consequently, we extracted [11C]-(+)-PHNO $BP_{ND}$ separately for the caudate and accumbens striatal sub-regions, in order to investigate the relationship between striatal D2/3R availability and task-induced cortical network connectivity in a more fine-grained manner. For completeness we also report results using D2/3R availability measured from the whole striatum (caudate, accumbens and putamen).

There was a positive correlation between caudate D2/3R availability and task-induced connectivity change in the DMN (rho = 0.45 [.18,. 65], d.f. = 46, P = 0.002), and DMN-TPN (rho = 0.33, [0.04, 0.57], d.f. = 46, P = 0.02), but not within the TPN (rho = 0.21, [-0.08, 0.48], d.f. = 46, P = 0.15). However, the difference between the correlations in the DMN and TPN was not itself significant (z = 1.41 [95% confidence interval for difference between correlations: −0.10,. 65], P = 0.16). D2/3R

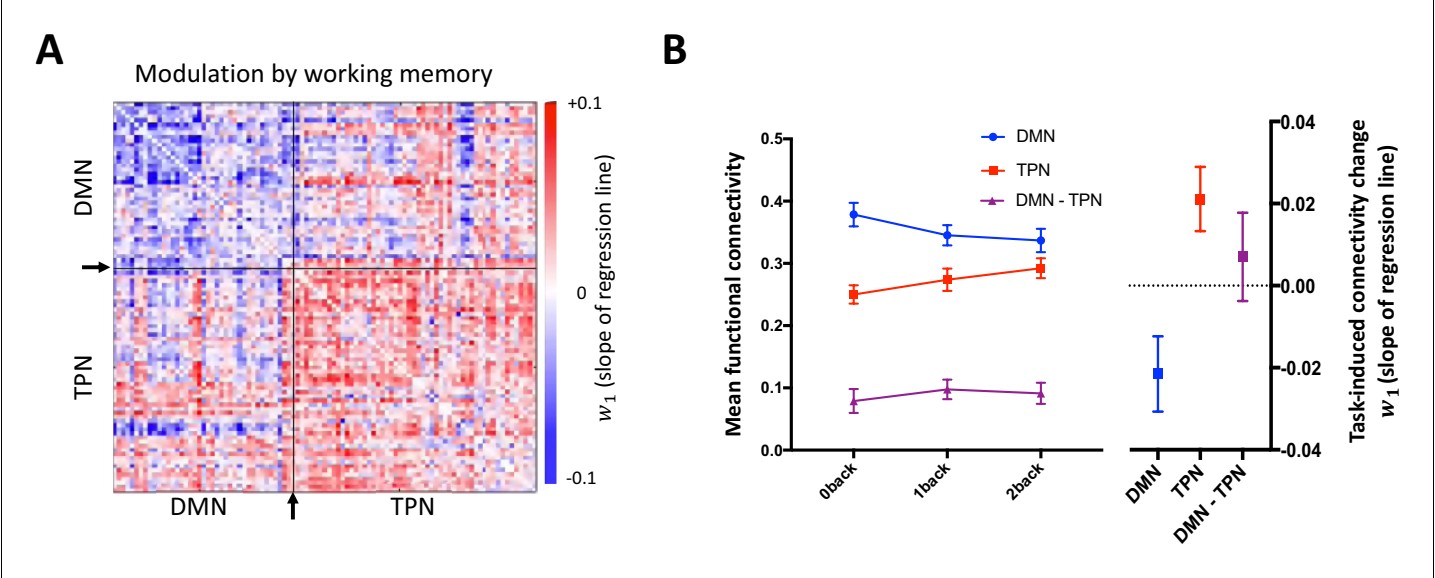

**Figure 3.** Task-induced changes in DMN and TPN functional connectivity. Increasing working memory load is accompanied by decreasing functional connectivity within the DMN (negative $w_1$ values) and increasing functional connectivity within the TPN (positive $w_1$ values). (**A**) Mean task-induced functional connectivity change of each network edge (node-node connection) in the whole sample. Each edge (cell of the matrix) represents the regression coefficient ($w_1$ value) of connectivity change as a function of working memory load, averaged over the whole group. Arrows and black lines indicate the boundary separating nodes allocated to the empirical DMN vs TPN. (**B**) Mean task-induced functional connectivity change within the DMN, TPN and DMN-TPN across the whole group. *Left:* Group mean (± S.E.M) functional connectivity strength (Fisher z-transformed r-value) within the DMN, TPN and DMN-TPN as a function of working memory load. *Right:* Group mean (± S.E.M) task-induced functional connectivity change ($w_1$) within the DMN, TPN and DMN-TPN. $w_1$ was significantly different to zero in both the DMN ($t_{50} = -2.33$, $P = 0.02$, one-sample t-test) and TPN ($t_{50} = 2.70$, $P = 0.01$, one-sample t-test), but not the DMN-TPN ($t_{50} = 0.65$, $P = 0.52$, one-sample t-test). Repeated measures ANOVA indicated that $w_1$ was not equal within the DMN, TPN and DMN-TPN edges ($F_{2,100} = 11.16$ $P < 0.001$), and post-hoc paired t-tests confirmed that the $w_1$ of the DMN was significantly lower than both the TPN and DMN-TPN ($t_{50} = -4.56$, $P < 0.001$ and $t_{50} = -3.01$, $P = 0.004$, respectively), but that there was no significant difference between the $w_1$ of the TPN and DMN-TPN ($t_{50} = 1.62$, $P = 0.11$).

DOI: https://doi.org/10.7554/eLife.45045.005

availability in the accumbens was similarly related to task-induced connectivity change in the DMN (rho = 0.31 [.017,. 55], d.f. = 46, P = 0.03), but not the TPN (rho = 0.003, [−0.29,. 29], d.f. = 46, P = 0.98) or DMN-TPN (rho = 0.15, [−0.15,. 42], d.f. = 46, P = 0.32). Although the magnitude of the D2/3R-DMN connectivity relationship was greater in the caudate compared to the accumbens, the difference between these two correlations was not statistically significant (z = 1.51 [-0.05, 0.37], P = 0.13).

Taking the striatum as a whole, there was a positive correlation between D2/3R availability in the whole striatum and the task-induced connectivity change within the DMN (rho = 0.38 [.10,. 60], d. f. = 46, P = 0.008), but not within the TPN (rho = 0.11 [−0.18,. 39], d.f. = 46, P = 0.44) or DMN-TPN (rho = 0.17 [−0.12,. 43], d.f. = 46, P = 0.29) (*Figure 4a*). This indicates that participants who exhibited greater task-induced reduction in DMN connectivity had lower striatal D2/3R availability, particularly in the caudate.

We investigated the specificity of the relationship between task-induced DMN connectivity change and striatal D2/3Rs in two additional analyses. First, there was no correlation between striatal D2/3R availability and functional connectivity averaged over the whole task session (i.e. mean network functional connectivity over all task blocks) in the DMN (Caudate: rho = 0.07 [−0.23,. 36], d. f. = 46, P = 0.63; Accumbens: rho = 0.05 [−0.25,. 34], d.f. = 46, P = 0.75; Whole Striatum: rho = 0.16 [−0.14,. 43], d.f. = 46, P = 0.29). Similarly, there was also no significant relationship between regional D2/3R availability and whole-session connectivity in the TPN or DMN-TPN (all P > 0.15). Second, there was no relationship between D2/3R availability in the substantia nigra/ventral tegmental area (SN/VTA, where the D3R fraction of the PHNO signal approaches 100% [*Tziortzi et al., 2011*]) and cortical network connectivity change (DMN: rho = 0.01 [−0.28,. 30], d.

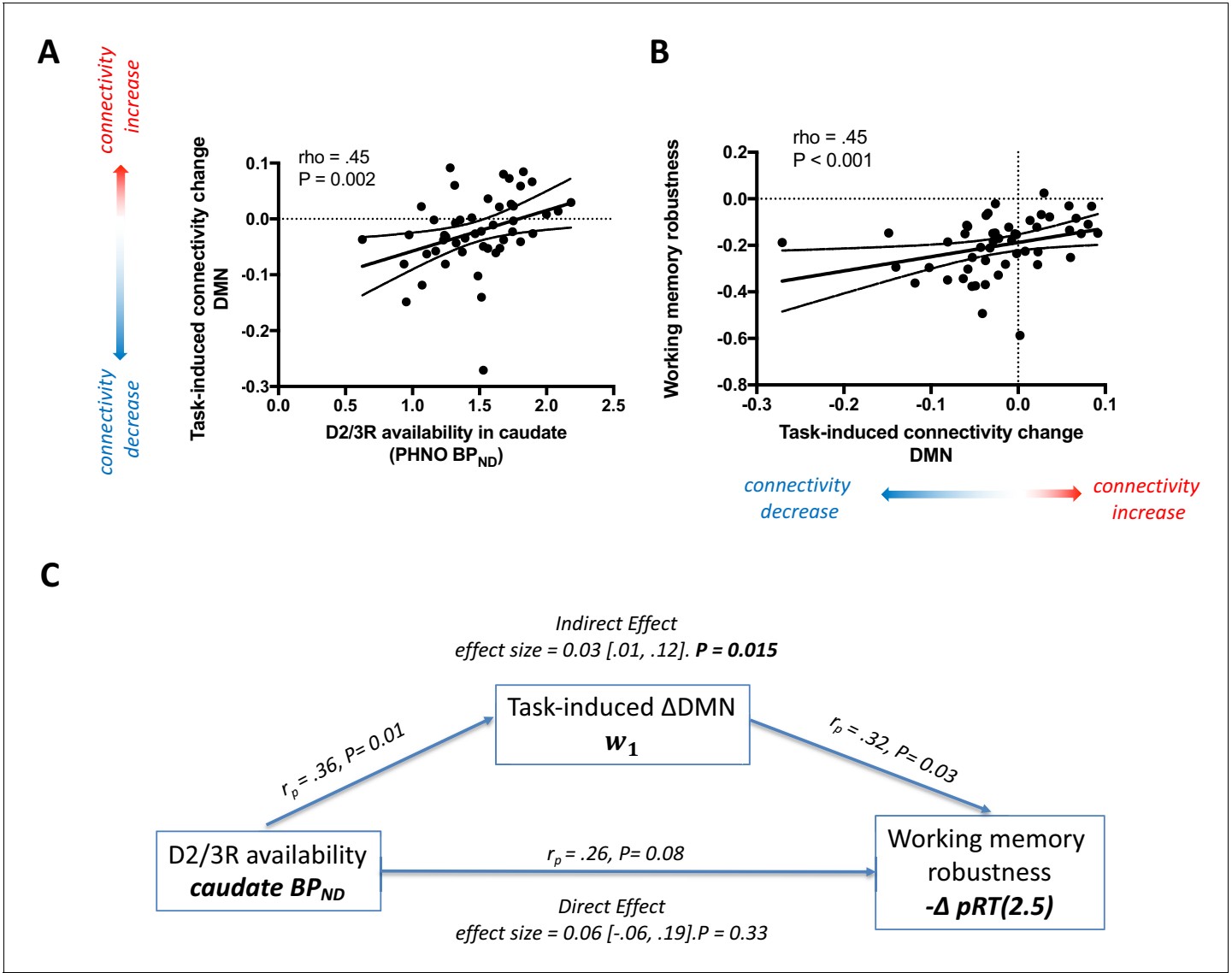

**Figure 4.** The relationship between striatal D2/3R availability, task-induced change in DMN connectivity, and task performance. (A) The significant positive correlation between striatal D2/3R availability and task-induced change in connectivity within the default mode network (DMN), where negative values on the y-axis indicate a reduction in connectivity. Lower caudate D2/3R availability is related to a task-induced reduction in DMN connectivity. (B) The relationship between task-induced functional connectivity change within the DMN and working memory robustness, $-\Delta\mathrm{pRT}(2.5)$ (a value of 0 indicates that performance is unaffected by increasing working memory load, while negative values indicate that performance decreases with increasing cognitive demands). Negative values on the x-axis indicate a task-induced reduction in DMN connectivity. Task-induced decreases in DMN connectivity were associated with greater working-memory related behavioural impairment. (C) Mediation analysis. Greater caudate D2/3R availability has a significant but indirect association with improved task performance, mediated via task-induced connectivity change within the DMN. Pearson's correlation coefficients in (C) are reported in the sample of 48 participants (d.f. = 46) who had both PET and fMRI.

DOI: https://doi.org/10.7554/eLife.45045.006

f. = 46, P = 0.93); TPN: (rho = −0.10 [−0.38,. 20], d.f. = 46, P = 0.50); DMN-TPN: (rho = −0.12 [−0.40,. 18], d.f. = 46, P = 0.42).

Of note, the correlation between DMN connectivity change and D2/3R availability in the SN/VTA was significantly weaker than the correlation with D2/3R availability in the caudate (z = 2.54 [0.11, 0.83], P = 0.01), and whole striatum (z = 2.29 [0.06, 0.72], P = 0.02), but not the ventral striatum (z = 1.7 [-0.05, 0.67], P = 0.09).

## The relationship between network connectivity changes and performance

There was a significant positive correlation between the task-induced change in DMN connectivity and working memory performance (defined as the robustness of performance to increasing cognitive load, $-\Delta pRT(2.5)$) (rho = .45 [.20,.64], d.f. = 49, P < 0.001), indicating that participants who showed the greatest task-induced reduction of functional connectivity within the DMN also showed the most marked load-dependent impairment in task performance (*Figure 4b*). This result was robust to a wide range of penalization ratios from 2.5 to 4 (all P < 0.001). There was no similar relationship for connectivity change within the TPN or DMN-TPN ($-\Delta pRT(2.5-4)$, all P > 0.50), and also no relationship between performance and mean functional connectivity averaged over the whole task session in the DMN or TPN ($-\Delta pRT(2.5-4)$, all P > 0.50).

## The relationship between striatal D2/3Rs and performance

There was a positive relationship between D2/3R availability in the caudate and $-\Delta pRT(2.5)$ (rho = 0.33 [.04,. 57], d.f. = 46, P = 0.02), which remained significant for penalization ratios from 2.5 to 4 (all P < 0.02). This relationship was not present for D2/3R availability in the accumbens ($-\Delta pRT(2.5)$: rho = 0.10 [−0.20,. 38], d.f. = 46,, P = 0.50. P > 0.4 for $-\Delta pRT(2.5-4)$), and the difference between these correlation coefficients was itself significant (z = 2.44 [0.05, 0.44], P = 0.01). The correlation between performance and D2/3R availability in the whole striatum was present at trend level ($-\Delta pRT(2.5)$: rho = 0.27 [−0.02,. 52], d.f. = 46, p=0.06. P < 0.06 for $-\Delta pRT(2.5-4)$). There was no relationship between D2/3R availability in the SN/VTA and performance ($-\Delta pRT(2.5)$ rho = 0.13 [−0.17,. 41], d.f. = 46, P = 0.37. P > 0.3 for $-\Delta pRT(2.5-4)$).

Striatal D2Rs are thought to exert an influence on cognitive function through a more direct effect on the excitability and function of cortical circuits (*Cools and D'Esposito, 2011*; *Frank and O'Reilly, 2006*; *Salami et al., 2019*). Having found a significant relationship between striatal D2/3R availability and both task-induced DMN connectivity, and a relationship between both measures and task performance, we tested whether the influence of striatal D2/3R availability on performance is mediated by the effect of the former on task-induced connectivity change within the DMN. A mediation analysis indicated that the effect of caudate D2/3R availability on working memory performance is mediated through the direct effect of D2/3R availability on task-induced DMN connectivity change ($-\Delta pRT(2.5)$: average causal mediation effect (ACME) = 0.03 [0.01, 0.12], P = 0.015. The mediation effect remains significant at P < 0.015 for $-\Delta pRT(2.5-4)$) (*Figure 4c*). This analysis was also significant using D2/3R availability from the whole striatum ($-\Delta pRT(2.5)$: ACME = 0.04 [0.01, 0.14], P = 0.026. P < 0.03 for $-\Delta pRT(2.5-4)$), but was not significant using D2/3R availability from the accumbens ($-\Delta pRT(2.5)$: ACME = 0.002 [<0.001, 0.12], P = 0.09. P > 0.08 for $-\Delta pRT(2.5-4)$). As expected, there was no significant mediation effect when the mediator variable was task-induced connectivity changes within the TPN (all P > 0.6) or DMN-TPN (all P > 0.3). These results suggest that striatal D2/3Rs, particularly in the caudate, might exert an influence on working memory performance through a more direct effect on task-induced functional connectivity changes within the DMN. Participants with higher striatal D2/3R availability exhibited the least pronounced DMN connectivity decreases with working memory load, which predicted more robust task performance in the face of increasing working memory load.

## Relationship between striatal D2/3R availability and task-related activation

Our results support the conclusion that striatal D2/3R availability, particularly within caudate, is related to task-induced changes in DMN connectivity and working memory performance. For completeness, we conducted two analyses to test for a relationship between D2/3R availability within this region and inter-individual differences in task-evoked BOLD activation. First, using a standard mass-univariate analysis we tested whether there was a relationship between the parametric BOLD response to working memory load and caudate D2/3R availability. We found no evidence for a linear or quadratic relationship at P < 0.05, whole-brain level. In a second analysis we used partial least squares (PLS) regression for fMRI, to test for the presence of a multivariate pattern of brain activity that shows a relationship to inter-individual differences in caudate D2/3R availability. This analysis again did not detect significant latent variables (LVs) showing a relationship to caudate D2/3R

availability (1st LV P-value = 0.20). However, consistent with the standard mass univariate fMRI analysis (*Figure 2a*), a standard PLS analysis found strong evidence for BOLD modulation by working memory load in a widespread cortical frontoparietal network (1st latent variable significant at P < 0.001).

## Within-subject relationship between DMN strength and performance

The between-subjects analysis so far has considered the relationship between the average task-induced connectivity change within the DMN and working memory performance, finding that as cognitive load increases, the individuals who showed the greatest DMN connectivity decreases also showed the greatest working memory performance degradation. We employed a within-subject linear regression approach to investigate whether block-specific DMN connectivity is a significant predictor of working memory performance (see Materials and methods). A simple regression analysis confirmed that block-wise DMN connectivity strength is a significant within-subject predictor of working-memory performance (using $pRT(2.5)$, $w_1$ = -.10 [-.17, -.02], P = 0.01). The results remain significant when using $pRT(2.5 - 4)$). When including block-specific working memory load as an additional predictor variable in the model, this effect was not statistically significant (using $pRT(2.5)$, $w_2$ = -.04 [-.10,. 01], P = 0.14. P > 0.2 when using $pRT(2.5 - 4)$). As expected, this analysis confirmed a significant effect of working memory load ($w_1$) on performance (all P < 0.001). As increased $pRT$ is indicative of poorer performance, the negative regression coefficients in these analyses indicate that in a given task block, stronger DMN connectivity is a predictor of improved working memory performance within an individual. This is in line with the between-subject finding that poorer performing participants show the most exaggerated DMN connectivity reductions with increased working memory load.

## Analysis using d-Prime behavioural measure

Finally, for completeness we repeated the analysis using the discriminability index (d') (*Haatveit et al., 2010*) as the measure of working memory performance. As with the $pRT$ metric, for each individual we regressed mean d' for each task condition on working memory load, and defined overall performance as the regression coefficient on working memory load. Using this performance metric, there was no relationship between task-induced DMN connectivity change and performance (rho = 0.12 [-.17,. 39], d.f. = 49, P = 0.4), and no relationship between performance and D2/3R availability in any striatal sub-region (all P > 0.2).

Importantly, compared to $pRT$, d' is not sensitive to differences in processing speed (i.e. reaction time) between task conditions with similar performance accuracy, and also cannot differentiate between (incorrect) response omissions and (correct) 'non-target' responses on non-target trials (the dominant trial type). Consistent with these theoretical considerations, in our sample d' was not significantly different between the two easiest task conditions (0-back d' = 3.5, 1-back d' = 3.6. P = 0.7), but was significantly worse in the 2-back condition (d' = 2.8) compared to both 0- and 1-back (both P < 0.001). In light of these results we conducted a series of exploratory analyses focussing on DMN functional connectivity and performance (d') changes from 0- to 1-back, and from 1- to 2-back separately. From the 1-back to 2-back condition there was a significant correlation between DMN connectivity change (DMN_{1back} − DMN_{2back}) and performance change (d'_{1back} − d'_{2back}) (rho = 0.30 [0.01, 0.54], d.f. = 49, P = 0.03). This relationship was not present for the 0-back minus 1-back contrast (rho = 0.19 [-.10,. 45], d.f. =49, P = 0.17). However, there was no relationship between performance change, either from 0- to 1-back, or from 1- to 2-back, and D2/3R availability in any striatal sub-region (all P > 0.29).

## Discussion

We measured striatal D2/3R availability and task-related cortical network connectivity in a large sample of young healthy participants. The novel contribution of the study is the demonstration that lower striatal D2/3R availability, particularly in the caudate, is associated with greater reduction in task-induced DMN functional connectivity, and that participants with greater reductions in DMN connectivity show worse performance with increasing working memory loads. Our findings indicate that caudate D2/3R availability might affect working memory performance by modulating task-induced functional connectivity changes within the DMN.

## Regional specificity of the relationship between D2/3R signalling and working memory

The primate striatum is anatomically organised in a functionally-specific manner, with ventral regions (e.g. accumbens) contributing to limbic and reward processing, and more dorsal regions (including dorsal caudate) supporting motor and executive (e.g. working memory) function (*Dahlin et al., 2008*; *Rieckmann et al., 2011*; *Levy et al., 1997*; *McCutcheon et al., 2019*; *Haber et al., 2000*). Several of our results are in accordance with this hypothesis. First, the correlation between striatal D2/3R availability and task-induced DMN connectivity change is numerically (but not significantly) greater in the caudate compared to the accumbens. Second, the positive correlation between striatal D2/3R availability and working memory performance was significantly greater in the caudate, compared with the accumbens. Third, the mediation analyses indicated a significant effect of caudate, but not accumbens, D2/3R availability on performance, mediated through a direct effect on DMN connectivity change. Furthermore, we report no significant relationships between D2/3R availability in the SN/VTA and DMN connectivity change or behaviour. Although these results are consistent with a regional functional specificity of D2/3R signalling within the mesostriatal circuit, their interpretation is complicated by the use of $[^{11}C]$-(+)-PHNO to measure D2/3R availability. $[^{11}C]$-(+)-PHNO is a strongly D3R-preferring agonist tracer, and there are large regional differences in D2R: D3R contribution to $[^{11}C]$-(+)-PHNO $BP_{ND}$. Specifically, the D3R fraction of this signal is negligible in the dorsal caudate, approximately 20–25% in the ventral striatum, and approaches 100% in the SN/VTA (*Tziortzi et al., 2011*; *Searle et al., 2010*). An alternative interpretation of our findings, therefore, is that there are significant differences in the manner in which mesostriatal D2R and D3R signalling supports working memory. Indeed, previous work has shown that D2R and D3R may have different effects on certain cognitive processes (*Groman et al., 2016*; *Groman et al., 2011*; *Groman et al., 2014*).

## The relationship between striatal D2/3R signalling, cognitive performance and cortical activity

We find a positive correlation between caudate D2R availability and working memory performance. A recent PET study in older adults (mean age 66.2) reported no relationship between caudate D2R availability (indexed using the D2R-preferring antagonist tracer $[^{11}C]$raclopride) and working memory performance, (*Nyberg et al., 2016*) suggesting that the relationship between D2/3R signalling and cognition may differ across the lifespan.

Theoretical and experimental studies implicate striatal dopamine, and particularly D2/3Rs, in the flexible updating of cortical representations, (*D'Esposito and Postle, 2015*; *Cools and D'Esposito, 2011*) possibly mediated via a disinhibition of thalamocortical loops (*Frank et al., 2001*).

We found a positive correlation between caudate D2/3R availability and task-induced cortical functional connectivity change within the DMN and the DMN-TPN, and a non-significant positive correlation with task-induced TPN connectivity. Of note, the difference between dopamine-DMN and dopamine-TPN correlations was not statistically significant, and we believe it is therefore premature to claim that the influence of D2/3R signalling on cortical excitability is specific for the DMN.

Recent animal studies shed light on the potential mechanisms through which striatal D2Rs might influence cortical network activity. Mice that have been genetically engineered to overexpress post-synaptic D2Rs in the striatum show impaired performance on tasks that require working memory updating and memory stabilisation in the face of interference (*Simpson and Kellendonk, 2017*). These animals also have increased D1R sensitivity and impaired inhibitory neurotransmission within the prefrontal cortex (PFC), (*Kellendonk et al., 2006*; *Li et al., 2011*) and manifest several abnormalities in dopaminergic ventral tegmental area (VTA) neuron function, including reduced mean firing, (*Krabbe et al., 2015*) reduced working memory-dependent phase-locking, and reduced working memory-dependent VTA-PFC synchrony (*Duvarci et al., 2018*). These changes may be due to an increase in inhibitory signalling from D2R-expressing striatal medium spiny neurons to the VTA (*Cazorla et al., 2012*). Interestingly, the reduction in task-related oscillatory coupling between VTA (but not substantia nigra) dopamine neurons and PFC is directly related to impaired working memory performance, (*Duvarci et al., 2018*) suggesting that our observed relationship between striatal D2/3R availability, cortical activity and task performance might reflect the influence of striatal D2/3Rs on mesocortical dopaminergic projections, via an inhibitory connection from the striatum to the

midbrain (*Krabbe et al., 2015*). However, it is unclear to what extent the neurophysiological alterations found in the D2R overexpression mouse model accurately mirror the effects of higher D2R availability in healthy human volunteers. Future animal experiments that employ PET, electrophysiological recordings, and fMRI measures of functional connectivity would help to address this question, and facilitate interpretation of human imaging results.

We find no evidence for a relationship between striatal D2/3R availability and task-induced BOLD activation using either mass-univariate or multivariate regression (PLS) approaches. We note with interest a recent [$^{11}$C]raclopride PET and fMRI study that used a PLS approach to investigate the relationship between working memory-induced BOLD activation and D2R availability in older adults (mean age 66.2) (*Salami et al., 2019*). The authors find that task-induced BOLD activation in widespread striatal and (largely frontal) cortical regions is related to D2R availability measures in a differential manner for normal- and low-performing participants. This study differs from the present study in several key respects. Age differences between the study samples suggest that the influence of dopamine on fMRI activation patterns may differ across the lifespan. Second, the study by Salami and colleagues used hierarchical factor modelling to estimate latent factors relating to striatal, neocortical and limbic D2/3R availability, and used these latent factors as the predictor variables in a single PLS model. Finally, differences in D2:D3R binding properties of [$^{11}$C]raclopride and [$^{11}$C]-(+)-PHNO may also contribute to the apparent discrepancy between the results.

## Beyond D2/3R signalling – the wider role of dopamine in cognitive performance and cortical activity

Beyond striatal D2/3R availability, previous human PET-fMRI studies have investigated the relationship between other facets of dopamine function and both cortical activity and performance during tasks. For example, PET studies investigating striatal dopamine synthesis capacity have reported positive, (*Berry et al., 2018a*; *Cools et al., 2008*; *Landau et al., 2009*; *Vernaleken et al., 2007*) negative, (*Braskie et al., 2008*; *Dang et al., 2012a*) and quadratic relationships (*Berry et al., 2016*) with performance on tasks requiring working memory and cognitive flexibility. This heterogeneity likely reflects the fact that optimal performance relies on a task-specific balance between stability and flexibility of neural representations, with the former supported by cortical D1Rs, and the latter supported in part by striatal D2Rs (*D'Esposito and Postle, 2015*; *Cools and D'Esposito, 2011*; *Frank et al., 2001*; *O'Reilly, 2006*; *Bilder et al., 2004*). Impaired performance can therefore arise in contexts of both excess (supra-optimal) and impoverished (sub-optimal) D2/3R signalling, (*Cools and D'Esposito, 2011*; *Frank and O'Reilly, 2006*; *Cools et al., 2007*; *Samanez-Larkin et al., 2013*) which may shed light on the nature of cognitive impairment in conditions of both elevated and reduced striatal dopamine signalling (*Millan et al., 2012*; *Braskie et al., 2008*; *Dahlin et al., 2008*; *Bäckman et al., 2000*; *Matuskey et al., 2016*; *Simpson et al., 2010*; *McCutcheon et al., 2018a*; *Meder et al., 2019*). It is therefore conceivable that the impaired working memory performance seen in psychiatric conditions such as schizophrenia reflects supra-optimal D2R signalling, whereas in our healthy volunteer population, non-pathological elevation in D2R availability was predictive of improved performance.

Neurally, striatal dopamine synthesis capacity has also been reported to have a quadratic relationship with task-induced frontoparietal activation, (*Berry et al., 2016*) with mixed results relating to its relationship with DMN deactivations (*Dang et al., 2012a*; *Braskie et al., 2011*). Moreover, two studies report that striatal D1R availability is related to frontoparietal connectivity averaged over an entire working memory task session, but not task-induced connectivity changes (*Rieckmann et al., 2011*; *Roffman et al., 2016*). However, we find no relationship between D2/3 receptor availability and whole-session functional connectivity within the DMN or TPN during the task.

Although there is significant covariation between striatal dopamine synthesis capacity and D2R availability, (*Berry et al., 2018b*) and between striatal D1R and D2R binding, (*Groman et al., 2014*) further dual-tracer human PET studies will be required to fully characterise the relationship between different facets of dopamine function reported in the discussed studies. It is noteworthy, however, that both theoretical and empirical studies suggest that different dopamine receptor subtypes have differential effects on working memory function and learning (*Groman et al., 2016*; *Groman et al., 2011*; *Groman et al., 2014*; *Durstewitz and Seamans, 2008*).

## The relationship between task-induced cortical connectivity and cognition

We report decreased DMN functional connectivity with increased working memory load, which is related to task performance. These results are consistent with studies that report task-induced decreases in DMN functional connectivity, and with studies that highlight the importance of cortical network reorganisation for task performance (*Shine and Poldrack, 2018*; *Shine et al., 2016*; *Cole et al., 2014*; *Finc et al., 2017*; *Liang et al., 2016*; *Vatansever et al., 2015*; *Fornito et al., 2012*). The relationship between task-induced decreases in DMN functional connectivity and impaired cognitive performance may reflect recent evidence that the DMN, rather than being 'task-negative' as such, is activated by conditions in which large shifts of attention are required during cognitive operations (*Smith et al., 2018*; *Arsenault et al., 2018*; *Crittenden et al., 2015*). Failure to recruit the DMN may therefore hamper the cognitively demanding selection and deselection of representations with increasing working memory loads.

## Limitations and strengths

We interpret our results in light of the prior literature showing that striatal dopamine may influence cognitive performance and task-induced neuronal activity. However, a key limitation of the present study is that it is not possible to make strong inferences regarding the direction of causality from purely observational studies. Related to this, there may be (unmeasured) neuronal variables, in addition to task-induced DMN connectivity change, which covary with striatal D2/3R availability, and which may exert a more direct causal influence on working memory performance. These concerns can only be addressed in fully randomised interventional experimental designs. Secondly, although our study does not allow us to test whether the reported relationships are specific to the D2/3R (or whether they additionally extend to the D1R, for example), previous studies have suggested that D2Rs and D1Rs have different relationships with cognitive performance and cortical connectivity (*Groman et al., 2014*; *Roffman et al., 2016*). Finally, the interpretation of the $[^{11}C]$-(+)-PHNO BP$_{ND}$ signal is inherently ambiguous, as it reflects both total D2/3R density, and endogenous synaptic dopamine levels. Future studies employing dopamine depletion interventions are therefore required to disambiguate which of these variables is more related to cortical connectivity and cognitive performance (*Caravaggio et al., 2014*). Our study has a number of strengths that extend the current literature, including the large sample size, measurement of D2/3R and cortical connectivity in the same participants, and examination of both task-induced and whole-session functional connectivity.

## Clinical implications

An improved mechanistic understanding of the relationship between striatal D2/3R signalling and cognitive performance is highly relevant for understanding the cognitive deficits seen in normal ageing (associated with reduced striatal D2R availability [*Bäckman et al., 2000*; *Matuskey et al., 2016*]), and neuropsychiatric conditions associated with dysregulated striatal dopamine such as Parkinson's disease and schizophrenia (*Millan et al., 2012*; *Simpson et al., 2010*; *McCutcheon et al., 2018a*; *Meder et al., 2019*). In schizophrenia, for example, there is elevated striatal dopamine signalling, (*McCutcheon et al., 2018a*) abnormal DMN resting functional connectivity and task-induced deactivation, (*Landin-Romero et al., 2015*; *Whitfield-Gabrieli et al., 2009*) and a well-replicated deficit in tasks involving working memory manipulation and resistance to distraction (*Simpson et al., 2010*; *Simpson and Kellendonk, 2017*; *Anticevic et al., 2012*). Moreover, animal evidence suggests that supra-optimal striatal D2R signalling may underlie some of these neurophysiological and cognitive abnormalities, providing one rationale for the use of D2/3R antagonists in symptomatic treatment. Our results shed new light on the potential neurophysiological and circuit mechanisms relating striatal D2/3R signalling to cognitive performance in healthy young adults, where inter-individual differences in receptor availability lie within the healthy population distribution. An unanswered question is whether the observed relationships between striatal D2/3R availability, task-induced cortical connectivity and task performance, are qualitatively similar to those seen in conditions such as schizophrenia or Parkinson's disease, or whether an inverted-U relationship is revealed when considering these clinical populations (*Cools and D'Esposito, 2011*; *Meder et al., 2019*). Future studies employing similar multimodal imaging designs in these clinical populations, ideally both on and off dopaminergic medication, will be crucial in addressing this question.

## Conclusions

We employ a multimodal neuroimaging approach to test the relationship between striatal D2/3R availability and working memory performance. We find that reduced D2/3R availability, particularly in the caudate, is directly related to a greater task induced reduction in DMN functional connectivity, and that greater reductions are associated with worse task performance. In a mediation analysis we show that D2/3R availability may support working memory performance through its relationship with task-induced DMN connectivity change. These findings could help explain natural variation in working memory performance, as well as cognitive decline seen in healthy ageing and cognitive impairments associated with neuropsychiatric disease.

# Materials and methods

## Participants

The study was approved by the local NHS Research Ethics Committee and the Administration of Radioactive Substances Advisory Committee, and was conducted at Imanova Centre for Imaging Sciences (Invicro Ltd), London. The initial study sample consisted of 53 participants, who all completed the n-back fMRI task session. Two participants were excluded from further analysis: one owing to MR imaging artefact caused by orthodontic braces, and a second owing to non-performance of the task ('no response' rates > 3 SD from the group mean). Fifty-one healthy participants were therefore included in the final fMRI analysis (mean age 25.8 years [SD 6.5], 30 male), forty-eight of whom had a $[^{11}C]$-(+)$-$4-propyl-9-hydroxy-naphthoxazine ($[^{11}C]$-(+)-PHNO) PET scan (two participants declined a PET scan, and a third terminated the scan due to nausea). All participants provided written informed consent to take part in the study.

## Task design and behavioural analysis

Details of the n-back task are shown in *Figure 1* and details of performance measurement are described in main text. The task was coded in PsychoPy version 2 (https://www.psychopy.org). Details of the proportional penalized reaction time (pRT) performance metric are presented in main text.

In addition to the pRT, we also present analyses using the discriminability index (d') performance metric. d' is defined as:

$$d' = Z(\text{hit rate}) - Z(\text{false alarm rate})$$

, where 'hit rate' is the proportion of targets that the participant correctly identified, 'false alarm rate' is the proportion of non-targets incorrectly identified as targets, and $Z(x)$ denotes the inverse of the cumulative distribution function of the Gaussian distribution. We adjusted perfect scores at each working memory load level with the following formulae: $1 - 1/2n$ for 'hit rate'=1, and $1/(2n)$ for 'false alarm rate'=0, where $n$ is the total number of targets (potential hits) or non-targets (potential false alarms) over the entire task (for 'hit rate' and 'false alarm rate', respectively) (*Haatveit et al., 2010*).

## MR image acquisition

Magnetic resonance (MR) images were acquired using a Siemens MAGNETOM Verio 3 T magnetic resonance imaging scanner and a 32-channel phased-array head-coil. We acquired a MPRAGE (Magnetization Prepared Rapid Gradient Echo) structural MRI scan from each participant at the start of the MRI scanning session, using parameters based on the Alzheimer's Disease Research Network (*Jack et al., 2008*) (ADNI-GO; 160 slices x 240 × 256, TR = 2300 ms, TE = 2.98 ms, flip angle = 9 degrees, 1 mm isotropic voxels, band-width = 240 Hz/pixel, parallel imaging (PI) factor = 2). We acquired B0 fieldmaps to unwarp EPI images during pre-processing (*Hutton et al., 2002*). Functional images were acquired with a multiband sequence, using a multiband acceleration factor of 2 (*Demetriou et al., 2018*). We acquired 275 whole-brain volumes per session, consisting of 72 interleaved slices (2 mm thickness), with a repetition time (TR) of 2000 ms, echo time (TE) of 30 ms, an in-plane resolution of 3 × 3 mm, flip angle of 62°, and bandwidth of 1906 Hz/pixel.

## fMRI activation analysis

### Mass-univariate analysis

We used Statistical Parametric Mapping 12 (SPM12) software (http://www.fil.ion.ucl.ac.uk/spm) for the event-related analysis. fMRI time series were realigned to the mean image, unwarped using the B0 fieldmaps generated by the Fieldmap toolbox, (*Hutton et al., 2002*) co-registered to the structural image and normalisation to MNI space using the DARTEL toolbox, (*Ashburner, 2007*) with 8 mm full-width at half-maximum (FWHM) Gaussian kernel smoothing. The first level general linear model (GLM) included a boxcar regressor for task blocks, accompanied by a parametric modulator for working memory load (*Roffman et al., 2016*; *Schmidt et al., 2009*), in addition to six motion realignment nuisance parameters. We additionally included temporal derivatives (first-order differences), applied an AR(1) model (to account for serial correlations in the fMRI time series) and used a 128 s cut-off high pass filter. At the second level we identified voxel clusters where blood-oxygen-level dependent (BOLD) signal was positively (activation) and negatively (deactivation) correlated with working memory load using one-sample t-tests on the estimated responses for the first level analysis. We used a second-level regression analysis to identify voxels that showed an activation/deactivation pattern that correlated with $-\Delta pRT(2.5)$ or $[^{11}C]$-(+)-PHNO BP$_{ND}$. Random field theory was used to correct to multiple comparisons. We report clusters that survive family-wise error correction at p<0.05 at the whole-brain cluster level (cluster forming threshold set to p<0.001 (uncorrected) to ensure a well behaved family error control) (*Eklund et al., 2016*).

### Spatiotemporal (ST-) Partial Least Squares (PLS) Regression analysis

In addition to the mass-univariate activation analysis, we also analysed the pre-processed functional images using the (multivariate) PLS approach, using the ST-PLS toolbox for MATLAB (https://www.rotman-baycrest.on.ca/pls/). First, we conducted a standard PLS analysis (with mean-centering) to identify brain regions where BOLD activity covaried with working memory load (where blocked conditions were: 'rest', '0back', 1-back' and '2back'). Next, we used 'Regular behavioural PLS' to investigate whether there were regions where the variation in BOLD in the task conditions was itself related to caudate D2/3R availability. This method is described in detail in *Salami et al. (2019)* and *McIntosh et al. (2004)*. In brief, the analysis conducts a singular value decomposition on the cross-block correlation matrix (correlation between BOLD activity across experimental conditions and dopamine measures) to identify orthogonal latent variables (LVs) that best represent the relationships between the BOLD activity and the task condition or dopamine measure (*Salami et al., 2019*). The statistical significance of each LV was assessed using permutation tests (n = 1000). LVs were considered statistically significant only if the proportion of permuted singular values that exceeded the original singular value was < 0.05.

## fMRI functional connectivity analysis

Our primary analysis focussed on the task-induced functional connectivity changes between cortical brain regions within task-positive and task-negative cortical networks. We define the functional connectivity between two brain regions (the strength of the 'edge' connecting two regions in a network, in the terminology of Graph Theory [*Rubinov and Sporns, 2010*]) as the Fisher z-transformed Pearson's correlation coefficient between the BOLD time-series in each region. Brain regions themselves (the 'nodes' of a network) are taken from the Gordon cortical parcellation, a network parcellation based on fMRI functional connectivity patterns observed in a sample of 120 healthy young adults (*Gordon et al., 2016*). The functional connectivity analysis pipeline consisted of (1) spatial and temporal pre-processing, (2) data-driven detection of empirical task-positive (TPN) and default mode (DMN) cortical networks, and (3) definition of mean load-dependent functional connectivity change within the TPN and DMN, similar to previous studies (*McCutcheon et al., 2018b*).

### Pre-processing

We employed a standard fMRI pre-processing pipeline implemented in the CONN toolbox (*Whitfield-Gabrieli and Nieto-Castanon, 2012*) (version 17 .f) (https://www.nitrc.org/projects/conn) within MATLAB. fMRI pre-processing included (1) slice timing correction, (2) realignment of functional scans, (3) normalisation to MNI space and (4) spatial smoothing (Gaussian kernel of 8 mm FWHM). In the denoising step we used linear regression to remove the influence of the following confounding

effects on the fMRI time course: (1) BOLD signal from the white matter and CSF voxels (five components each, derived using the anatomical component-based correction (aCompCor) implemented using the ART toolbox), (2) six residual head motion parameters and their first order temporal derivatives, (3) scrubbing of artefact/outlier scans (average intensity deviated more than five standard deviations from the mean intensity in the session, or composite head movement exceeded 0.9 mm from the previous image), and (4) effect of task-condition using separate block regressors (for 0-, 1-, and 2-back conditions) convolved with the haemodynamic response function. Thus, we performed the connectivity analysis on the residuals of the BOLD time series after removing condition-related activation/deactivation effects (*Finc et al., 2017*; *Whitfield-Gabrieli and Nieto-Castanano, 2012*; *Fair et al., 2007*). Finally, the denoising step included temporal bandpass filtering (0.008–0.09 Hz), and linear detrending of the functional time course. We did not included global signal regression of the grey matter voxels in the de-noising process as this is known to introduce spurious anti-correlations between large networks (*Murphy and Fox, 2017*). Following pre-processing, we performed condition-dependent functional connectivity analysis on the mean BOLD time course (*Whitfield-Gabrieli and Nieto-Castanano, 2012*) extracted from cortical nodes defined by the Gordon parcellation (*Gordon et al., 2016*).

## Task-positive and task-negative network detection

We used the functional connectivity measures for each working memory condition to generate an undirected weighted covariance matrix, $C$, for each cognitive load, where $c_{ij}$ represents the Fisher z-transformed Pearson's correlation coefficient between region $i$ and $j$ (i.e. edge strength between nodes).

We used the Louvain algorithm (implemented in the MATLAB Brain Connectivity Toolbox, *Rubinov and Sporns, 2010*; https://sites.google.com/site/bctnet/) to define group level default mode (DMN) and task-positive (TPN) networks, on the basis of the functional connectivity covariance matrix of each individual participant during the '0-back' condition (the implicit motor- and sustained attention-control condition), setting negative weights to 0 (*Cohen and D'Esposito, 2016*).

This Louvain algorithm partitions the nodes of the covariance matrix into a community structure that maximises the modularity of the network, where modularity (Q) is a measure of network segregation, and describes the degree to which a network may be subdivided into non-overlapping communities, with maximal number of within-community connections, and a minimum number of between-community connections (*Shine et al., 2016*; *Rubinov and Sporns, 2010*; *Finc et al., 2017*; *Rubinov and Sporns, 2011*). For positive weighted networks it is defined as:

$$Q = \frac{1}{v} \sum_{ij} \left( w_{ij} - \frac{s_i s_j}{v} \right) \delta_{M_i M_j},$$

where $v$ is the total strength of the network ($v = \sum_{ij} w_{ij}$), $w_{ij}$ is the strength of the edge connecting node $i$ to $j$, $s_i$ is the strength of node $i$ ($s_i = \sum_j w_{ij}$)), and $\delta$ is set to 1 when node $i$ and $j$ are in the same module, and 0 otherwise (*Finc et al., 2017*; *Rubinov and Sporns, 2011*; *Schultz and Cole, 2016*).

We restricted the community detection analysis *a priori* to the 97 nodes allocated to the default mode network (DMN), dorsal attention network (DAN) and frontoparietal network (FPN), as defined by the Gordon atlas, (*Gordon et al., 2016*) as these networks have been extensively studied in working memory paradigms and show a relationship to dopamine function (*Rieckmann et al., 2011*; *Liang et al., 2016*; *Roffman et al., 2016*; *Dang et al., 2012b*; *Spreng et al., 2010*; *Cassidy et al., 2016*).

Given the stochastic nature of the Louvain algorithm we used a consensus clustering approach to ensure the robustness of the final community structure (*Lancichinetti and Fortunato, 2012*; *Cohen and D'Esposito, 2016*; *Hearne et al., 2017*). Specifically, for each subject we iteratively applied the algorithm 1000 times with different initial random seeds. This generated 1000 separate partitions, which were then combined to a single agreement matrix, $D$, where entry $d_{ij}$ represents the proportion of partitions in which nodes $i$ and $j$ were assigned to the same community. Following a thresholding step (in which all entries <50% agreement were set to zero), (*Cohen and D'Esposito, 2016*) the agreement matrix was then subjected to another 1000 iterations of the Louvain algorithm. This procedure was repeated until each of the resulting 1000 partitions was equal, resulting in a

binary agreement community matrix (i.e. a consensus partition) for each participant. Finally, a mean group agreement matrix was created by averaging the consensus partition agreement matrices of each participant. This group matrix was then itself subjected to the consensus clustering procedure, until a final group community partition was produced. We used the default resolution parameter ($\gamma = 1$).

We used the Sørensen-Dice similarity coefficient (DSC) (*Dice, 1945*) to quantify the overlap between cortical voxels of the empirically-derived DMN and task positive network (TPN) nodes, and the voxels of the deactivation and activation clusters from our event related analysis,

$$DSC = \frac{2|X \cap Y|}{|X| + |Y|},$$

where X and Y are the two sets of cortical voxels to be compared and |X| and |Y| are the number of voxels in each set. We restricted the number of voxels in the activation/deactivation sets to those lying within a cortical grey matter mask (i.e. the intersection of a cerebral hemisphere mask from SPM12 PickAtlas and the voxels with grey matter tissue probability value of >50% from the SPM12 tissue probability map). We used a similar analysis to evaluate the overlap between cortical nodes assigned to the empirical (Louvain algorithm) and *a priori* (Gordon atlas) task-relevant networks.

## Load-dependent functional connectivity definition

For each participant and each edge (connection between nodes of the cortical covariance matrix), we fitted a simple linear regression model of functional connectivity (Fisher z-transformed r-value) as a function of working memory load, $\mathrm{zFC} = \mathrm{w_0} + \mathrm{w_1} * \mathrm{WM_{load}}$, and used the $w_1$ regression coefficient as a measure of load-dependent connectivity change for that edge. For each individual a summary statistic for task-induced connectivity change within the TPN and DMN was defined as the mean $w_1$ value of all edges within the network. We contrast this task-induced functional connectivity change with a measure of functional connectivity averaged over the whole task session (both analyses include identical pre-processing steps) (*Whitfield-Gabrieli and Nieto-Castanon, 2012*).

## Within-subject analysis of relationship between DMN connectivity and performance

To investigate the influence of cortical network connectivity on working memory performance within individuals, for each individual we first regressed block-specific performance (*pRT$_i$*) against block-specific DMN strength:

$$pRT_i(2.5) = w_0 + w_1 * DMN\_strength_i$$

(where i$\in$ *{1:18}* designate the 18 task blocks). We also repeated this analysis using a model that contained block-specific working memory load as an additional independent variable:

$$pRT_i(2.5) = w_0 + w_1 * WM\_load_i + w_2 * DMN\_strength_i$$

To evaluate the significance of the effect of block-specific DMN connectivity on *pRT$_i$*(2.5) at the group level we used a one-sample t-test on the regression coefficient weighting the *DMN_strength* term.

## PET image acquisition and analysis

[$^{11}$C]-(+)-PHNO positron emission tomography (PET) images were acquired using a Siemens Biograph HiRez XVI PET scanner. Each subject first received a low-dose computed tomography scan for attenuation and model-based scatter correction before the injection of a single intravenous bolus of 0.020–0.029 micrograms/kg [$^{11}$C]-(+)-PHNO (a high affinity D2/3R agonist ligand) (*Narendran et al., 2006*). The mean [$^{11}$C]-(+)-PHNO mass administered was 1.5 micrograms (SD 0.31) and mean injected activity was 177.5 MBq (SD 50.0). Dynamic emission data were acquired continuously for 90 min.

Dynamic PET images were reconstructed using a filtered back-projection algorithm into 31 frames ($8 \times 15$ s, $3 \times 60$ s, $5 \times 120$ s, $15 \times 300$ s) with a 128 matrix, a zoom of 2.6 and a transaxial Gaussian filter of 5 mm. For PET image analysis we employed an automatic pipeline to obtain an individual

parcellation of the brain, implemented in MIAKAT (release 4.2.6, http://www.miakat.org), (*Gunn et al., 2016*) SPM12 and FSL (version 5.0.9, https://fsl.fmrib.ox.ac.uk/fsl/fslwiki).

Structural MR images were segmented using SPM12 functions to obtain grey matter masks used for the definition of the reference region during the kinetic analysis. The ICBM152 structural brain template was non-linearly warped to each subject's structural MR image, and the derived deformation parameters were applied to our neuroanatomical atlas to obtain a parcellation of each subject's brain into the studied anatomical regions of interest (bilateral whole striatum (caudate, putamen and accumbens), accumbens, and caudate, as defined by the CIC atlas, [*Tziortzi et al., 2011*] and substantia nigra/ventral tegmental area [*Nour et al., 2018*]). The MRI, associated individual parcellation and associated grey matter masks were then down-sampled to the PET resolution (2 mm). Dynamic PET images were corrected for motion using a frame-by-frame registration process with a mutual information cost function. For each subject the averaged PET image from the entire scan duration was registered to the down-sampled structural MRI scan with rigid-body registration. The rigid body matrix was subsequently applied to the motion corrected dynamic PET. Regional time activity curves (TAC) were obtained by applying the down-sampled individual anatomical parcellations to the motion-corrected dynamic PET image.

The non-displaceable binding potential ($BP_{ND}$) of [$^{11}$C]-(+)-PHNO is defined as follows:

$$BP_{ND} = \frac{f_{ND} B_{avail}}{K_D}$$

where $f_{ND}$ is the non-displaceable free fraction of PHNO in the brain, $B_{avail}$ is the number of D2/3Rs available to be bound by the radioligand, and $1/K_D$ is the affinity of radioligand for the target. The simplified reference tissue model (SRTM) was used to derive $BP_{ND}$ from the regional time activity curves (*Gunn et al., 1997*; *Lammertsma and Hume, 1996*). We used cerebellar grey matter as the reference region, defined as the intersection of the warped cerebellum atlas (*Tziortzi et al., 2011*) and individual subject grey matter mask.

## Statistical Analysis

Statistical analysis was performed using MATLAB (R2017a) and R (version 3.3.1). $P < 0.05$ (two tailed) was considered statistically significant for the primary analyses. We hypothesised that the DMN and TPN would show opposite task-induced connectivity changes with increasing working memory load, and tested this hypothesis using repeated measures ANOVA and *post hoc* t-tests. Subsequently, we used Spearman's rank correlation coefficient to test for (direct) monotonic relationships between task-induced connectivity changes within cortical networks and both striatal D2/3R availability ([$^{11}$C]-(+)-PHNO $BP_{ND}$) and task performance (where, for each participant, task-induced connectivity change within a network was defined as the mean $w_1$ value for all edges in the network). Spearman's rank coefficient was used as we did not assume that the monotonic relationship between variables would be linear. We tested the hypothesis that striatal D2/3R availability influences working memory task performance by modulating cortical network connectivity with a mediation analysis using the 'mediation' package in R, implementing a nonparametric bootstrap method with bias-corrected and accelerated confidence intervals and 10,000 simulation draws. We used the *cocor* package in R (http://comparingcorrelations.org/) to compare overlapping, dependent, correlation (Spearman's rank) coefficients at $P < 0.05$ (2-tailed), using Meng et al.'s method for comparing correlation coefficients (including 95% confidence intervals for difference between correlation coefficients) (*Diedenhofen and Musch, 2015*; *Meng et al., 1992*).

## Acknowledgements

MN is a pre-doctoral fellow of the International Max Planck Research School on Computational Methods in Psychiatry and Ageing Research (IMPRS COMP2PSYCH). The participating institutions are the Max Planck Institute for Human Development, Berlin, Germany, and University College London, London, UK. For more information, see: https://www.mps-ucl-centre.mpg.de/en/comp2psych. MN is also supported by the National institute for Health Research UK and Wellcome Trust. TD is supported by an EU-FP7 MC6 ITN IN-SENS grant (grant number 607616). RM is supported by the Wellcome Trust (no. 200102/Z/15/Z). RA is supported by the Academy of Medical Sciences (AMS-SGCL13-Adams) and the National Institute of Health Research (CL-2013-18-003). This study was

funded by Medical Research Council-UK (no. MC-A656-5QD30), and Wellcome Trust (no. 094849/Z/10/Z) grants to OH and the National Institute for Health Research Biomedical Research Centre at South London and Maudsley NHS Foundation Trust and King's College London. The views expressed are those of the authors and not necessarily those of the NHS, funding bodies, or the Department of Health. We thank the MR and PET technologists and radiographers at Imanova Centre for Imaging Sciences (Invicro Ltd), London, and Christopher Coello for help with the PET image analysis.

## Additional information

### Competing interests

Matthew B Wall: is affiliated with Imanova Centre for Imaging Sciences (Invicro Ltd). The author has no other competing interests to declare. The other authors declare that no competing interests exist.

### Funding

| Funder | Grant reference number | Author |
| --- | --- | --- |
| National Institute for Health Research | Academic Clinical Fellowship | Matthew M Nour |
| Wellcome | 200102/Z/15/Z | Robert A McCutcheon |
| Academy of Medical Sciences | AMS-SGCL13-Adams | Rick A Adams |
| Medical Research Council | MC-A656-5QD30 | Oliver D Howes |
| Wellcome | 094849/Z/10/Z | Oliver D Howes |
| Seventh Framework Programme | EU-FP7 MC6 ITN IN-SENS 607616 | Tarik Dahoun |
| Wellcome | UCL Wellcome Trust PhD Programme for Clinicians | Matthew M Nour |

The funders had no role in study design, data collection and interpretation, or the decision to submit the work for publication.

### Author contributions

Matthew M Nour, Conceptualization, Formal analysis, Investigation, Visualization, Methodology, Writing—original draft, Project administration, Writing—review and editing; Tarik Dahoun, Conceptualization, Formal analysis, Investigation, Methodology, Project administration; Robert A McCutcheon, Formal analysis, Methodology, Writing—review and editing; Rick A Adams, Investigation, Methodology, Project administration, Writing—review and editing; Matthew B Wall, Conceptualization, Software, Methodology, Writing—review and editing; Oliver D Howes, Conceptualization, Supervision, Funding acquisition, Project administration, Writing—review and editing

### Author ORCIDs

Matthew M Nour https://orcid.org/0000-0003-0858-6184
Robert A McCutcheon http://orcid.org/0000-0003-1102-2566
Rick A Adams http://orcid.org/0000-0002-7661-8881
Oliver D Howes https://orcid.org/0000-0002-2928-1972

### Ethics

Human subjects: The study was approved by the local NHS Research Ethics Committee (REC number: 12/LO/1955) and the UK Administration of Radioactive Substances Advisory Committee. All participants provided written informed consent to take part in the study, and for the results to be published.

Decision letter and Author response
Decision letter https://doi.org/10.7554/eLife.45045.009
Author response https://doi.org/10.7554/eLife.45045.010

## Additional files

### Supplementary files
• Transparent reporting form
DOI: https://doi.org/10.7554/eLife.45045.007

### Data availability

Neuroimaging summary data and analysis scripts (MATLAB and R) are publicly available on GitHub (https://github.com/matthewnour/D2_nback_connectivity_eLife2019; copy archived at https://github.com/elifesciences-publications/D2_nback_connectivity_eLife2019).

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
