## [Decision Letter]

Thank you for submitting your article "Task-induced functional brain connectivity mediates the relationship between striatal D2/3 receptors and working memory" for consideration by *eLife*. Your article has been reviewed by two peer reviewers and the evaluation has been overseen by David Badre as Reviewing Editor and Michael Frank as the Senior Editor. The reviewers have opted to remain anonymous.

The reviewers have discussed the reviews with one another and the Reviewing Editor has drafted this decision based on those reviews to help you prepare a revised submission.

Summary:

Working memory is thought to depend on interactions among functional networks, and is also known to be modulated by striatal dopamine mechanisms. However, the relationship among these factors and working memory performance is largely unknown. This paper uses a combination of fMRI and PET to probe interactions between working memory load, brain network activity, and D2/D3R availability. The study reports that the relationship between D2/D3R availability and behavior is mediated by default mode network activity, specifically. This is a novel observation in this domain with potentially wide ranging application.

Essential revisions:

The reviewers were positive overall about the paper. They both thought it makes an important and novel contribution to an important problem in the working memory literature. However, the reviewers also felt that there were a number of points in the analyses and interpretation of results that must be revised to strengthen the paper and clarify its contribution. I will summarize these here, largely using the reviewers' own language, as they have been clear and incisive in their comments.

1) Inferential complications introduced by the use of [^11^C]-(+)-PHNO (hereafer "PHNO")

To the authors' credit, they frequently (though not uniformly) use the phrase "D2/D3R" to reflect the ambiguity about whether principle PHNO's molecular target is actually D2Rs or D3Rs. At the same time, this phrase elides potentially key distinctions between PHNO vs. more traditional antagonist tracers like raclopride, as well as potentially key distinctions between the molecular correlate of striatal vs. SN/VTA PHNO BP_ND_. First, PHNO is a strongly D3-preferring agonist tracer, unlike the more modestly D2-preferring antagonist tracers used in most prior literature on this topic. This difference is material: the D3 receptor has at least 20-fold higher affinity than D2 for dopamine; PHNO has 4-48 fold higher affinity for D3 than for D2 (see Graff-Guerero et al., 2009 for review); and, most of the dorsal striatal PHNO signal may be D2-related, whereas most of the SN/VTA signal may be D3-related (Matuskey et al., 2016). Consequently:

1a) The apparent specificity of the association between striatal "D2/D3R" availability and load-induced DMN change in functional connectivity would be predicted either due to a differential role of striatum in this association, or due to a differential role of D2Rs over D3Rs in this association, or both. In other words, the biological feature which drives this apparent specificity is fundamentally ambiguous.

1b) This specificity may be more apparent than real. That is, the association of load-induced DMN FC change with striatal PHNO BP_ND_ is not directly compared with the association of load-induced DMN FC change with SN/VTA PHNO BP_ND_. A significant difference between these associations (i.e., a direct statistical comparison) would be required to claim specificity.

1c) One potential reason for the lack of specificity is the use of a whole-striatum ROI. A whole-striatum ROI may confound the D3R-dominant PHNO signal from ventral striatum with the D2R-dominant PHNO signal from dorsal striatum. In addition, and as the authors note, dorsal (associative) striatum is most theoretically relevant for the hypotheses being addressed here. Thus, use of a whole-striatum mask seems inappropriately coarse given what is known about PHNO; more specific masks (e.g., separating dorsal/associative from ventral/limbic striatum) would seem better justified both theoretically and biologically.

2) Inferential complications introduced by the treatment of inter-individual differences

Besides PHNO, another aspect of the authors' methods complicates the interpretation of the results: the near exclusive use of (between-)individual differences. This motivates two specific concerns:

2a) Simpsons paradox.

One tacit assumption of the manuscript is that Simpson's paradox does NOT occur in this domain. In other words, the same mechanisms that govern variance between-individuals (e.g., relating load-induced functional connectivity change in the DMN to load-induced behavioral performance change) are also operative within individuals, or at least that these variables are related with the same sign. Because of the pseudorandomized block-wise administration of the task, the authors have an fantastic opportunity to test this tacit assumption, at least in part. Does block-by-block variation in each subject's mean pRT score also predict block-by-block variation in DMN FC strength?

2b) Use of standardized beta coefficients.

In the mediation model (see also point 3 below) the authors use the standardized beta coefficient to quantify task-induced change in intrinsic DMN functional connectivity. This is an interesting choice, because it amounts to eliminating between-individual differences in range of intrinsic DMN functional connectivity observed, and normalizing all to have a variance of 1. Why is this the assumption made, i.e., why is the range of intrinsic DMN functional connectivity not relevant to striatal PHNO BP_ND_? The use of standardized betas seems to be a theoretically-laden choice (though whether it makes any difference is an interesting empirical question that the authors might choose to answer, if they did not intend to take a theoretical stance here).

3) The use of mediation analyses

Independent of the above issues, the use of mediation analysis introduces some further interpretational ambiguities to the extent that specific assumptions are not met. Two specific assumptions seem particularly relevant here:

3a) The authors note that a total effect is unnecessary for a mediation effect to be interpreted, *if* there is good reason to believe a true causal effect exists (here they refer Hayes, 2009). But what are the independent reasons to expect a causal effect of striatal BP_ND_ on behavioral performance at the between-individual level, and what evidence is there to justify the assumption? (See here also the point about Simpson's paradox, above).

3b) One assumption related to the use of mediation analysis is that the mediator is independent of all unmeasured variables that affect the outcome – an assumption that rarely holds outside of designs in which the mediator is experimentally randomized (Green, Ha, and Bullock, 2010). Perhaps the authors could speak to whether their use of mediation analysis verifiably meets this and other assumptions, and any consequences to the interpretations they offer here.

4) The current findings should be more actively interpreted within the context of previous dopamine PET studies that have already demonstrated relationships between striatal dopamine and functional connectivity during memory/working memory/executive function tasks (e.g. Rieckmann et al., 2011; Klostermann et al., 2012; Berry et al., 2018; Rieckmann et al., 2019 see also Stelzel et al., 2010; Stelzel et al., 2013 for genetic/pharmacological evidence), as well as previous studies reporting that striatal dopamine's influence on cognition is mediated by functional connectivity (Nyberg et al., 2016; Berry et al., 2018). There simply isn't enough known about how different dopamine PET measures are related within individuals to make strong claims (e.g. "the relationship between these mechanisms has not been tested in humans") that D2/3 findings are telling us something uniquely different from what transporter or synthesis findings have told us (see Berry et al., 2018 for within-subject correlation between D2/3 availability and synthesis capacity). The authors will find that these previous studies have focused on task positive networks and have largely examined connectivity between striatal ROIs and cortex. Thus, the focus on large-scale networks and DMN in particular should be emphasized as the major contribution of the current study. It would be to the advantage of the authors to use the Introduction and Discussion to adequately frame this finding. What is the role of DMN connectivity changes during challenge? Coupling/uncoupling? Are there specific changes in task-related DMN connectivity in aging or schizophrenia?

5) Does the relationship with functional connectivity and the mediation hold true for d' measures? What about% change in RT measures? I'm concerned that the effects reported here rely on replacing omissions and error responses with 2s RT entries.

6) The authors use a whole striatal ROI for PET analyses (relatedly, see comment 1C above), but caudate/dorsal caudate are more strongly implicated in working memory functions. Please demonstrate the mediation (and null results) do not rely on use of whole striatal ROI rather than caudate or dorsal caudate.

7) Salami and colleagues (2019) published a PET-fMRI study with a very similar design "Dopamine D2/3 binding potential modulates neural signatures of working memory in a load-dependent fashion." They do find relationships between D2/3 receptor availability and multivariate patterns of BOLD using a PLS approach. These finding are in agreement with other studies demonstrating BOLD-PET relationships. I don't think this study can be ignored. I recommend the authors apply PLS analysis to their dataset. If the results are still negative, this would potentially reflect interesting age-related effects and strengthen their claims that the effects of D2/3 receptor availability are truly limited to changes in DMN connectivity. If this new analysis approach reveals BOLD-PET relationships, this would serve as a worthwhile replication.

8) The authors evoke dopamine's modulation of fronto-striato-thalamic loops as a mechanism underlying its influence on cognition. How do these loops play into dopamine's modulation of task *negative* networks?

---

## [Author Response]

Essential revisions:1) Inferential complications introduced by the use of [^11^C]-(+)-PHNO (hereafer "PHNO")

*To the authors' credit, they frequently (though not uniformly) use the phrase "D2/D3R" to reflect the ambiguity about whether principle PHNO's molecular target is actually D2Rs or D3Rs. At the same time, this phrase elides potentially key distinctions between PHNO vs. more traditional antagonist tracers like raclopride, as well as potentially key distinctions between the molecular correlate of striatal vs. SN/VTA PHNO* BP_ND_*. First, PHNO is a strongly D3-preferring agonist tracer, unlike the more modestly D2-preferring antagonist tracers used in most prior literature on this topic. This difference is material: the D3 receptor has at least 20-fold higher affinity than D2 for dopamine; PHNO has 4-48 fold higher affinity for D3 than for D2 (see Graff-Guerero et al., 2009 for review); and, most of the dorsal striatal PHNO signal may be D2-related, whereas most of the SN/VTA signal may be D3-related (Matuskey et al., 2016). Consequently:*

1a) The apparent specificity of the association between striatal "D2/D3R" availability and load-induced DMN change in functional connectivity would be predicted either due to a differential role of striatum in this association, or due to a differential role of D2Rs over D3Rs in this association, or both. In other words, the biological feature which drives this apparent specificity is fundamentally ambiguous.1b) This specificity may be more apparent than real. That is, the association of load-induced DMN FC change with striatal PHNO BP_ND_ is not directly compared with the association of load-induced DMN FC change with SN/VTA PHNO BPND. A significant difference between these associations (i.e., a direct statistical comparison) would be required to claim specificity.1c) One potential reason for the lack of specificity is the use of a whole-striatum ROI. A whole-striatum ROI may confound the D3R-dominant PHNO signal from ventral striatum with the D2R-dominant PHNO signal from dorsal striatum. In addition, and as the authors note, dorsal (associative) striatum is most theoretically-relevant for the hypotheses being addressed here. Thus, use of a whole-striatum mask seems inappropriately coarse given what is known about PHNO; more specific masks (e.g., separating dorsal/associative from ventral/limbic striatum) would seem better justified both theoretically and biologically.

We thank the reviewers for raising these important points. Throughout the results we have adopted the suggestion to report results separately for striatal regions where the PHNO BP_ND_ has negligible D3R-fraction (caudate) and significant D3R fraction (accumbens). We additionally report results for whole striatum BP_ND_ for completeness.

“[^11^C]-(+)-PHNO binds selectively to both D2 and D3 receptors, and the D3R fraction of this measure differs between the ventral and dorsal striatum. In the (dorsal) caudate, the D3R fraction of the [^11^C]-(+)-PHNO BP_ND_ signal is negligible, while in the ventral striatum (including accumbens) the D3R fraction has been estimated at 20-25%.^32^ […] For completeness we also report results using D2/3R availability measured from the whole striatum (caudate, accumbens and putamen).”

Furthermore, we have included additional analyses that speak to the specificity of the relationship between striatal D2/3R availability and DMN connectivity. First, the magnitude of the correlation between caudate D2R availability and DMN connectivity change was numerically (although not significantly) greater than the relationship between accumbens D2/3R availability and DMN connectivity.

“There was a positive correlation between caudate D2/3R availability and task-induced connectivity change in the DMN (rho = .45 [.18,.65], d.f. = 46, P = 0.002), and DMN-TPN (rho = 0.33, [0.04, 0.57], d.f. = 46, P = 0.02), but not within the TPN (rho = 0.21, [-0.08, 0.48], d.f. = 46, P = 0.15). […] Although the magnitude of the D2/3R-DMN connectivity relationship was greater in the caudate compared to the accumbens, the difference between these two correlations was not statistically significant (z = 1.51 [-0.05, 0.37], P = 0.13).”

Second, we formally test whether the association between striatal (particularly caudate) D2/3R availability and DMN connectivity is significantly greater than that between SN/VTA D3R availability and DMN connectivity, finding this to be the case.

“Of note, the correlation between DMN connectivity change and D2/3R availability in the SN/VTA was significantly weaker than the correlation with D2/3R availability in the caudate (z = 2.54 [0.11, 0.83], P = 0.01,) and whole striatum (z = 2.29 [0.06, 0.72], P = 0.02), but not the ventral striatum (z = 1.7 [-0.05, 0.67], P = 0.09).”

We agree with the reviewers that the interpretation of these findings is not trivial. The stronger association between DMN connectivity change and caudate, as opposed to accumbens/SNVTA BPND, may reflect a genuine anatomical specificity, or may be an artefact of the larger D3R fraction in the PHNO signal for ventral striatum/SNVTA. We include this caveat in the Discussion.

***“***Regional specificity of the relationship between D2/3R signalling and working memory

The primate striatum is anatomically organised in a functionally-specific manner, with ventral regions (e.g. accumbens) contributing to limbic and reward processing, and more dorsal regions (including dorsal caudate) supporting motor and executive (e.g. working memory) function.[…]Indeed, previous work has shown that D2R and D3R may have different effects on certain cognitive processes.^39–41^”

2) Inferential complications introduced by the treatment of inter-individual differencesBesides PHNO, another aspect of the authors' methods complicates the interpretation of the results: the near exclusive use of (between-)individual differences. This motivates two specific concerns:2a) Simpsons paradox.One tacit assumption of the manuscript is that Simpson's paradox does NOT occur in this domain. In other words, the same mechanisms that govern variance between-individuals (e.g., relating load-induced functional connectivity change in the DMN to load-induced behavioral performance change) are also operative within individuals, or at least that these variables are related with the same sign. Because of the pseudorandomized block-wise administration of the task, the authors have a fantastic opportunity to test this tacit assumption, at least in part. Does block-by-block variation in each subject's mean pRT score also predict block-by-block variation in DMN FC strength?

We thank the reviewers for this suggestion. As requested, we have conducted a within-subject regression analysis to investigate whether mean DMN connectivity strength predicts behavioural performance in a given task block. These results are broadly consistent with the main between-individual findings, and argue against the presence of Simpson’s paradox in our data.

**“**Within-subject relationship between DMN strength and performance

The between-subjects analysis so far has considered the relationship between the average task-induced connectivity change within the DMN and working memory performance, finding that as cognitive load increases, the individuals who showed the greatest DMN connectivity decreases also showed the greatest working memory performance degradation. […] This is in line with the between-subject finding that poorer performing participants show the most exaggerated DMN connectivity reductions with increased working memory load.”

2b) Use of standardized beta coefficients.

*In the mediation model (see also point 3 below) the authors use the standardized beta coefficient to quantify task-induced change in intrinsic DMN functional connectivity. This is an interesting choice, because it amounts to eliminating between-individual differences in range of intrinsic DMN functional connectivity observed, and normalizing all to have a variance of 1. Why is this the assumption made, i.e., why is the range of intrinsic DMN functional connectivity not relevant to striatal PHNO* BP_ND_*? The use of standardized betas seems to be a theoretically-laden choice (though whether it makes any difference is an interesting empirical question that the authors might choose to answer, if they did not intend to take a theoretical stance here).*

We would like to clarify that we do not use standardized beta coefficients in any regression model in the paper, and we agree with the reviewers that the use of standardized regression coefficients would eliminate interesting between-individual differences in the range of intrinsic DMN connectivity.

We suspect that the misunderstanding on this point stemmed from our use of unclear terminology on two counts. First, in the original manuscript we described regression coefficients as ‘betas’. We have changed this to ‘w’ [weight] in the revised manuscript. Second, we have clarified the terminology ‘z-transformed Pearson’s r-value’ to ‘Fisher z-transformed Pearson’s r-value’. We hope that this clarifies our methodology.

3) The use of mediation analysesIndependent of the above issues, the use of mediation analysis introduces some further interpretational ambiguities to the extent that specific assumptions are not met. Two specific assumptions seem particularly relevant here:3a) The authors note that a total effect is unnecessary for a mediation effect to be interpreted, if there is good reason to believe a true causal effect exists (here they refer Hayes, 2009). But what are the independent reasons to expect a causal effect of striatal BPND on behavioral performance at the between-individual level, and what evidence is there to justify the assumption? (See here also the point about Simpson's paradox, above).

We thank the reviewer for these points.

There is good evidence that D2/3R signalling is important for cognitive function, including working memory. Specifically, impaired performance can arise in contexts of both excess and impoverished D2/3R signalling, a hypothesis supported by a wealth of pharmacological studies(Cools, Sheridan, Jacobs and D’Esposito, 2007; Cools and D’Esposito, 2011; Frank and O’Reilly, 2006; Samanez-Larkin et al., 2013). Similar to previous studies, our implicit assumption is that the effect of neuromodulatory signalling (e.g. dopamine) on cognition manifests through a more direct effect of the neuromodulator on the neuronal circuits that support the cognitive processes. We have made this explicit in the revised manuscript, in the results paragraph that introduces the mediation analysis:

“Striatal D2Rs are thought to exert an influence on cognitive function through a more direct effect on the excitability and function of cortical circuits.(Cools and D’Esposito, 2011; Frank and O’Reilly, 2006; Salami et al., 2019)”

Additionally, in the revised manuscript we do indeed detect a total effect of caudate D2/3R availability on working memory performance, which may address the concern raised in this comment more directly.

Concerns regarding Simpson’s paradox are addressed in response to comment 2.

3b) One assumption related to the use of mediation analysis is that the mediator is independent of all unmeasured variables that affect the outcome – an assumption that rarely holds outside of designs in which the mediator is experimentally randomized (Green, Ha, and Bullock, 2010). Perhaps the authors could speak to whether their use of mediation analysis verifiably meets this and other assumptions, and any consequences to the interpretations they offer here.

We have included a sentence in the Limitations section regarding the inferences that may be made using mediation analyses outside of experimentally randomized contexts.

“… a key limitation of the present study is that it is not possible to make strong inferences regarding the direction of causality from purely observational studies. Related to this, there may be (unmeasured) neuronal variables, in addition to task-induced DMN connectivity, which covary with striatal D2/3R availability, and which may exert a more direct causal influence on working memory performance. These concerns may only be addressed in fully randomized interventional experimental designs.”

4) The current findings should be more actively interpreted within the context of previous dopamine PET studies that have already demonstrated relationships between striatal dopamine and functional connectivity during memory/working memory/executive function tasks (e.g. Rieckmann et al., 2011; Klostermann et al., 2012; Berry et al., 2018; Rieckmann et al., 2019 see also Stelzel et al., 2010; Stelzel et al., 2013 for genetic/pharmacological evidence), as well as previous studies reporting that striatal dopamine's influence on cognition is mediated by functional connectivity (Nyberg et al., 2016; Berry et al., 2018).

We thank the reviewers for these suggestions. We fully agree that there is a vast literature on the relationship between dopamine, task-induced fMRI activation and cognitive performance. In the revised Discussion we actively interpret our results in light of human PET studies that have investigated the relationship between striatal D2/3Rs and working memory performance or task-induced cortical activity/connectivity.

“We find a positive correlation between caudate D2R availability and working memory performance. A recent PET study in older adults (mean age 66.2) reported no relationship between caudate D2R availability (indexed using the D2R-preferring antagonist tracer [^11^C]raclopride) and working memory performance,(Nyberg et al., 2016) suggesting that the relationship between D2/3R signalling and cognition may differ across the lifespan[…] Finally, differences in D2R:D3R binding properties of [^11^C]raclopride and [^11^C]-(+)-PHNO may also contribute to the apparent discrepancy between the results.”

We also dedicate a large section of the Discussion to human PET studies that investigate the relationship between other facets of dopamine (dopamine synthesis capacity, D1R availability) and task performance/neural activity.

However, we are concerned that a more extensive discussion of the wider literature relating (non-D2/3R) dopamine signalling to cognitive performance and cortical activity will overburden an already very long paper and Discussion section.

***“***Beyond D2/3R signalling – the wider role of dopamine in cognitive performance and cortical activity

Beyond striatal D2/3R availability, previous human PET-fMRI studies have investigated the relationship between other facets of dopamine function and both cortical activity and performance during tasks. […] This highlights that findings from PET studies measuring striatal dopamine synthesis capacity are limited in their ability to inform on the specific cognitive or neurophysiological roles of dopamine receptor subtypes, and that D1Rs and D2Rs within the striatum likely serve different functional roles.”

There simply isn't enough known about how different dopamine PET measures are related within individuals to make strong claims (e.g. "the relationship between these mechanisms has not been tested in humans") that D2/3 findings are telling us something uniquely different from what transporter or synthesis findings have told us (see Berry et al., 2018 for within-subject correlation between D2/3 availability and synthesis capacity).

We acknowledge this point, and agree that the current lack of human multi-tracer PET studies limits our ability to make definitive claims about how the PHNO BP_ND_ signal covaries with other dopamine PET measures (e.g. striatal dopamine synthesis capacity as measured by F-DOPA, D1R availability, DAT binding). However, we would also like to highlight that both theoretical and empirical studies suggest different functional roles for dopamine receptor subtypes, and that in certain neuropsychiatric conditions there may be abnormalities in one facet of dopamine signalling, but not another (e.g. in schizophrenia where there is a large elevation in striatal dopamine synthesis capacity, with equivocal changes in D1R and D2R binding). We have included the following paragraph in the Discussion.

“Although there is significant covariation between striatal dopamine synthesis capacity and D2R availability,(Berry, Shah, Furman, et al., 2018) and between striatal D1R and D2R binding,(Groman et al., 2014) further dual-tracer human PET studies will be required to fully characterise the relationship between different facets of dopamine function reported in the discussed studies. It is noteworthy, however, that both theoretical and empirical studies suggest that different dopamine receptor subtypes have differential effects on working memory function and learning.(Durstewitz and Seamans, 2008; Groman et al., 2014; Groman et al., 2011, 2016)”

We have deleted the clause “…however the relationship between these mechanisms has not been tested in humans.” in the Abstract.

The authors will find that these previous studies have focused on task positive networks and have largely examined connectivity between striatal ROIs and cortex. Thus, the focus on large scale networks and DMN in particular should be emphasized as the major contribution of the current study. It would be to the advantage of the authors to use the Introduction and Discussion to adequately frame this finding.

We have highlighted the novel contribution of our study in the Introduction and Discussion:

Introduction:

“Together, these findings suggest that striatal D2/3R levels, particularly within the caudate/dorsal striatum, might exert an influence on working memory performance by modulating task-induced functional connectivity within task-relevant cortical networks. However, this hypothesis has not been tested in humans. In this study we address this question directly…”

Discussion:

“The novel contribution of the study is the demonstration that lower striatal D2/3R availability, particularly in the caudate/dorsal striatum, is associated with greater reduction in task-induced DMN functional connectivity, and that participants with greater reductions in DMN connectivity show worse performance with increasing working memory loads.”

What is the role of DMN connectivity changes during challenge? Coupling/uncoupling?

The following sentence in the Discussion addresses this point:

“[Our] results are consistent with studies that report task-induced decreases in DMN functional connectivity, and with studies that highlight the importance of cortical network reorganisation for task performance.(Cole, Bassett, Power, Braver and Petersen, 2014; Finc et al., 2017; Fornito, Harrison, Zalesky and Simons, 2012; Liang, Zou, He and Yang, 2016; Shine et al., 2016; Shine and Poldrack, 2017; Vatansever, Menon, Manktelow, Sahakian and Stamatakis, 2015)”

Are there specific changes in task-related DMN connectivity in aging or schizophrenia?

To the best of our knowledge there are no studies investigating *task-induced changes* in DMN connectivity in schizophrenia. Numerous studies report abnormal connectivity during rest, blunted deactivation of DMN regions (e.g. mPFC) during tasks, and effective connectivity abnormalities. We briefly mention these in the Clinical Implications section:

“In schizophrenia, for example, there is elevated striatal dopamine signalling,(Robert McCutcheon et al., 2017) abnormal DMN resting functional connectivity and task-induced deactivation,(Landin-Romero et al., 2015; Whitfield-Gabrieli et al., 2009) and a well-replicated deficit in tasks involving working memory manipulation and resistance to distraction.(Anticevic, Repovs, Krystal and Barch, 2012; Simpson and Kellendonk, 2017; Simpson et al., 2010)”

However, none of these findings speak directly to the core question of the manuscript, and the focus of the manuscript is not on neuropsychiatric disorders. As such, we believe it would overburden the discussion to open a discussion on more general DMN abnormalities in ageing and neuropsychiatric disorders.

5) Does the relationship with functional connectivity and the mediation hold true for d' measures? What about% change in RT measures? I'm concerned that the effects reported here rely on replacing omissions and error responses with 2s RT entries.

Our decision to use the penalized reaction time (please also see response to Minor Point 2) as a performance metric was motivated primarily by 2 reasons. First, signal detection metrics, such as d’, are unable to differentiate between performance on conditions where accuracy (hit rate / false positive rate) is matched, but where processing time is different. Second, as d’ is defined as a contrast between the ‘hit rate’ and ‘false positive rate’, this measure cannot differentiate between (incorrect) response omissions and (correct) ‘non-target’ responses on non-target trials (the dominant trial type).

Nevertheless, we agree with the reviewers that it is of general interest to report the more equivocal results for the widely used d’ performance metric.

**“Analysis using d-Prime behavioural measure**

Finally, for completeness we repeated the analysis using the discriminability index (d’)(Haatveit, Sundet, Hugdahl, Ueland and Andreassen, 2010) as the measure of working memory performance. […] However, there was no relationship between performance change, either from 0- to 1-back, or from 1- to 2-back, and D2/3R availability in any striatal sub-region (all P > 0.29).”

6) The authors use a whole striatal ROI for PET analyses (relatedly, see comment 1C above), but caudate/dorsal caudate are more strongly implicated in working memory functions. Please demonstrate the mediation (and null results) do not rely on use of whole striatal ROI rather than caudate or dorsal caudate.

As explained in our response to point 1C, above, on receiving the reviewer’s comments we have moved towards presenting all results relating to D2/3R availability separately for the caudate, accumbens and whole striatum. All results presented in the original manuscript using the whole striatum ROI are also present when using the caudate ROI. Specifically, regarding the mediation results:

“A mediation analysis indicated that the effect of caudate D2/3R availability on working memory performance is mediated through the direct effect of D2/3R availability on task-induced DMN connectivity change (ΔpRT(2.5): average causal mediation effect (ACME) = 0.03 [0.01, 0.12], P = 0.015. Mediation remains significant at P < 0.015 for -ΔpRT(2.5-4))(Figure 4C). […]These results suggest that striatal D2/3Rs, particularly in the caudate, might exert an influence on working memory performance through a more direct effect on task-induced functional connectivity changes within the DMN.”

7) Salami and colleagues (2019) published a PET-fMRI study with a very similar design "Dopamine D2/3 binding potential modulates neural signatures of working memory in a load-dependent fashion." They do find relationships between D2/3 receptor availability and multivariate patterns of BOLD using a PLS approach. These finding are in agreement with other studies demonstrating BOLD-PET relationships. I don't think this study can be ignored. I recommend the authors apply PLS analysis to their dataset. If the results are still negative, this would potentially reflect interesting age-related effects and strengthen their claims that the effects of D2/3 receptor availability are truly limited to changes in DMN connectivity. If this new analysis approach reveals BOLD-PET relationships, this would serve as a worthwhile replication.

We thank the reviewers for bringing this study to our attention, which was published after our initial submission. We agree that the results are highly relevant, and we have conducted the suggested (‘behavioural’) PLS analyses on our data, to test whether the relationship between caudate D2R availability and task-induced BOLD signal extends to activation patterns as well as connectivity changes. We have included the PLS analysis results in a Resultssection that addresses this question directly.

**“**Relationship between striatal D2/3R availability and task-related activation.

Our results support the conclusion that striatal D2/3R availability, particularly within caudate, is related to task-induced changes in DMN connectivity and working memory performance. […]However, consistent with the standard mass univariate fMRI analysis (Figure 2A), a standard PLS analysis found strong evidence for BOLD modulation by working memory load in a widespread cortical frontoparietal network (1^st^ latent variable significant at P < 0.001).”

We do not replicate the findings of Salami and colleagues, and in the discussion cite several potential reasons for this, including (1) age differences between cohorts, (2) D2/3R measure differences (i.e. the use of a hierarchical factor model to estimate the striatal D2/3R factor in Salami et al), and (3) tracer differences (use of raclopride in Salami et al).

“We find no evidence for a relationship between striatal D2/3R availability and task-induced BOLD activation using either mass-univariate or multivariate regression (PLS) approaches. […] Finally, differences in D2R:D3R binding properties of [^11^C]raclopride and [^11^C]-(+)-PHNO may also contribute to the apparent discrepancy between the results.”

8) The authors evoke dopamine's modulation of fronto-striato-thalamic loops as a mechanism underlying its influence on cognition. How do these loops play into dopamine's modulation of task negative networks?

We agree with the reviewer that it is somewhat surprising that the influence of striatal D2/3R availability was expressed predominantly in the association with DMN connectivity change. We note that in the revised manuscript there are also significant correlations between caudate D2/3R availability and task-induced connectivity change between nodes of the DMN and TPN, but there is no relationship to behaviour. Any interpretation of the specific relationship between striatal D2/3R signalling and DMN vs TPN is highly speculative (and may be premature given our DA-fMRI correlations). We acknowledge this in the discussion:

“We found a positive correlation between caudate D2/3R availability and task-induced cortical functional connectivity within the DMN and the DMN-TPN, and a non-significant positive correlation with task-induced TPN connectivity. Of note, the difference between dopamine-DMN and dopamine-TPN correlations was not statistically significant, and we believe it is therefore premature to claim that the influence of D2/3R signalling on cortical excitability is specific for the DMN.”